# Scalable magnetoreceptive e-skin for energy-efficient high-resolution interaction towards undisturbed extended reality

Pavlo Makushko[1,6], Jin Ge [1,6] ✉, Gilbert Santiago Cañón Bermúdez[1], Oleksii Volkov [1], Yevhen Zabila[1], Stanislav Avdoshenko[2], Rico Illing[1], Leonid Ionov[3], Martin Kaltenbrunner [4,5], Jürgen Fassbender [1], Rui Xu [1] ✉ & Denys Makarov [1] ✉

Electronic skins (e-skins) seek to go beyond the natural human perception, e.g., by creating magnetoperception to sense and interact with omnipresent magnetic fields. However, realizing magnetoreceptive e-skin with spatially continuous sensing over large areas is challenging due to increase in power consumption with increasing sensing resolution. Here, by incorporating the giant magnetoresistance effect and electrical resistance tomography, we achieve continuous sensing of magnetic fields across an area of $120 \times 120$ mm$^2$ with a sensing resolution of better than 1 mm. Our approach enables magnetoreceptors with three orders of magnitude less energy consumption compared to state-of-the-art transistor-based magnetosensitive matrices. A simplified circuit configuration results in optical transparency, mechanical compliance, and vapor/liquid permeability, consequently permitting its imperceptible integration onto skins. Ultimately, these achievements pave the way for exceptional applications, including magnetoreceptive e-skin capable of undisturbed recognition of fine-grained gesture and a magnetoreceptive contact lens permitting touchless interaction.

Driven by substantial advancements in extended reality (XR), the internet of things (IoTs), and artificial intelligence (AI), the distinction between the physical and digital worlds continuously become fluid as their convergence grows[1,2]. An intuitive and responsive interface is urgently needed to effectively bridge the gap between these two realms and better communicate information with humans. This can be achieved in the form of electronic skins (e-skins) which are artificially designed to help humans interact with their environment by enhancing human perception or recovering lost sensory organs[3–6]. An ideal e-skin is expected to possess and combine the following features: (I)

preserving essential physiological functionalities of human skins such as mechanical compliance (stretchability, flexibility)[7,8], the ability to reflect visual cues[9,10], and permeability to liquid/vapor[11]; (II) capable of continuous operation across extensive surfaces without spatial interruption[12]; (III) high-resolution sensing for precise interaction[13]; (IV) energy-efficient operation over large areas to ensure prolonged service[14]; (V) remaining performance even in noisy or disruptive environments, e.g., in high-sweat and/or covered by cloth[15,16]; (VI) mitigating infections and transmission of bacteria/viruses, especially concerning health related or long-term applications[17]. Numerous

[1]Helmholtz-Zentrum Dresden-Rossendorf e.V., Institute of Ion Beam Physics and Materials Research, Bautzner Landstrasse 400, 01328 Dresden, Germany. [2]Institute for Solid State Research, Leibniz Institute for Solid State and Materials Research Dresden, 01069 Dresden, Germany. [3]Faculty of Engineering Science, Biofabrication, University of Bayreuth, Ludwig-Thoma-Str. 36a, 95447 Bayreuth, Germany. [4]Division of Soft Matter Physics, Institute for Experimental Physics, Johannes Kepler University, Altenberger Str. 69, 4040 Linz, Austria. [5]Soft Materials Lab, Linz Institute of Technology, Johannes Kepler University, Altenberger Str. 69, 4040 Linz, Austria. [6]These authors contributed equally: Pavlo Makushko, Jin Ge. ✉e-mail: gejin@mail.sysu.edu.cn; r.xu@hzdr.de; d.makarov@hzdr.de

examples of functional e-skins, which are sensitive to temperature, pressure, and light among other stimuli, have been demonstrated[18–20]. Despite these fascinating achievements, incorporating all of the above features into one e-skin remains elusive.

Magnetoreception, the ability to perceive external magnetic fields that is widely used by numerous species for orientation and navigation, has been recently added to the repertoire of skin-like sensory organs. This includes skin compliant giant magnetoresistive (GMR)[21,22], magnetic tunnel junction (MTJ)[23], anisotropic magnetoresistive (AMR)[24,25], spin valves[26,27], magnetoimpedance[28], and Hall sensors[29], serving as a tool for studying cognition mechanisms and opening a new channel for interactivity (Fig. 1a). Thanks to the touchless characteristic of magnetic interaction and its immunity to diverse disruptions in daily life, these magnetoreceptive elements open the door for development and integration of undisturbed (Feature V) and hygienic (Feature VI) e-skins. Nevertheless, challenges remain, for example, achieving spatially continuous and seamless sensing over large areas (Features II) is far beyond the capabilities of these sensor individuals, as is the optimization in sensing resolution (Feature III) and energy consumption (Feature IV). To overcome some of these obstacles, magnetosensitive active matrices were recently demonstrated that employ arrays of discrete sensing units composed of magnetoresistive

sensors and thin film organic transistors (Fig. 1b)[30,31]. Yet, such approach is not readily scalable towards expansive on-skin interface since the structural complexity (amount of circuit elements and electrode contacts) and the power consumption dramatically increase with the interactive area. Furthermore, the complex and multilayered circuit layouts inevitably impair one or more physiological attributes of (human) skin such as its mechanical, visual, and permeable aspects (Feature I). Therefore, it is imperative to develop a system that enables skin-like magnetoreception capable of tracking magnetic stimuli over large areas with only a few circuit elements and minimal power demands.

When confronted with challenges in creating efficient configurations for functions, nature often showcases its distinctive yet remarkably graceful solutions[32–35]. Inspired by spiders and eels that sense local variation over large areas using only a simple sensing medium (i.e., webs and surrounding water) and a limited number of signal channels (e.g., spider legs and electroreceptors distributed bilaterally along the eel's body), we propose a magnetoreceptor, composing a single magnetoreceptive layer contacted with 16 (8 or 32 is also feasible) measurement electrodes (Fig. 1c). Enabled by an incorporation of the giant magnetoresistance effect and electrical resistance tomography, our magnetoreceptor achieves large-area continuous sensing of

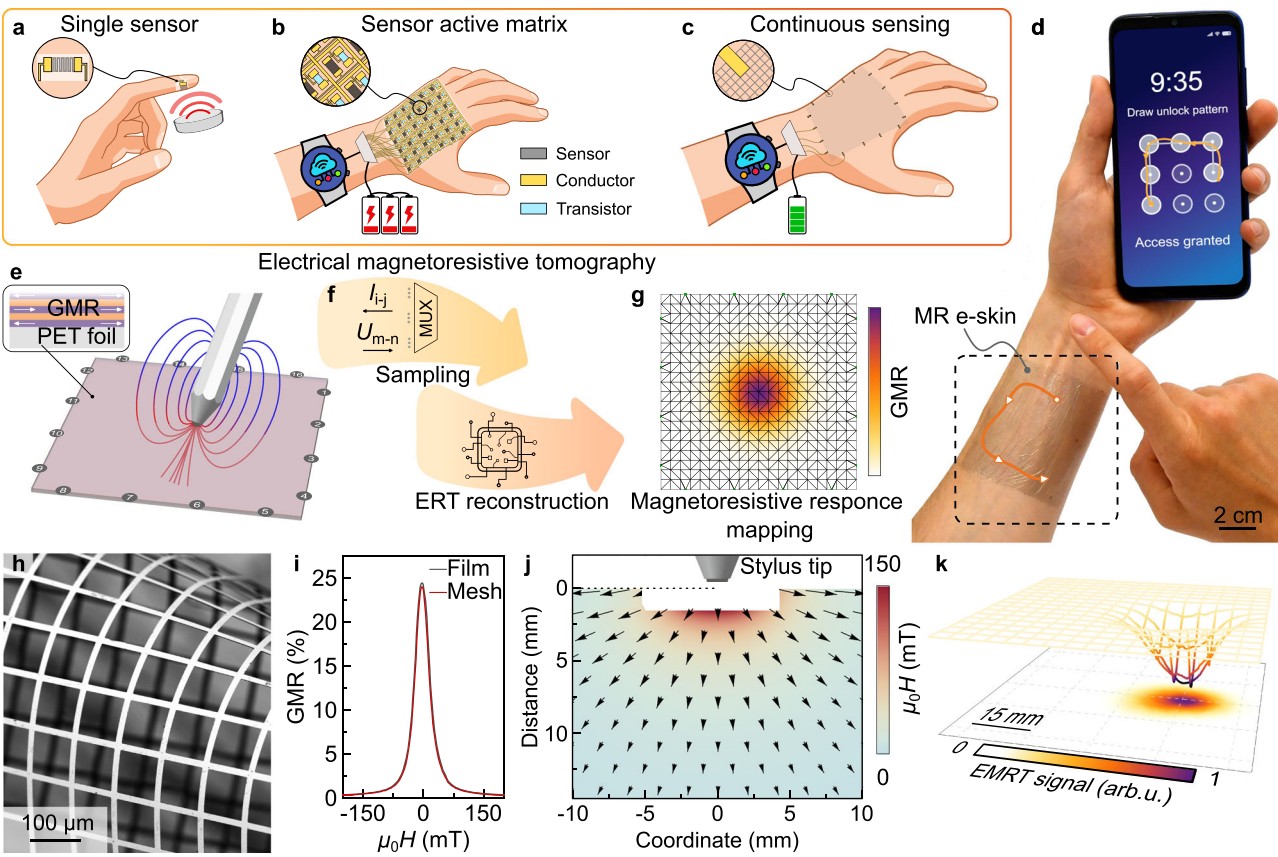

**Fig. 1 | Evolution of magnetoreceptive e-skins towards large area interaction. a** Magnetosensitive flexible sensing element. **b** On-skin active matrix of magnetic field sensors for large-area distributed sensing sequentially probed using a multiplexer (MUX). **c** EMRT-enabled magnetoreceptive e-skin for spatially continuous sensing over large areas. **d** Magnetoreceptive e-skin probed by EMRT algorithm and adhered onto a wrist of a person is used for interaction with a smartphone. A concept figure depicting human-machine interaction shown in Supplementary Movie 1 and Supplementary Fig. 9. **e**–**g** The mechanism of EMRT: (**e**) continuous magnetoreceptive layer (here, GMR medium) with measurement electrodes distributed along its perimeter. **f** The sequentially probed and the recorded voltage data (indices $i,j$ stand for electrode pair to supply current and $m,n$ for pair on which

voltage is measured) is fed into tomography reconstruction algorithm that returns (**g**) the distribution of giant magnetoresistive (GMR) induced resistivity change in the sensing medium. **h** A representative SEM image of a perforated magnetoreceptive mesh. The fabrication protocol was repeated for 5 times and displays consistently reproducible result. **i** GMR characteristic of the magnetoreceptive medium on PET foil. **j** Stray field map around the tip of a magnetic stylus, used for the interaction with the EMRT-enabled magnetoreceptor. Colorcoded background represents the magnetic flux $\mu_0 H$ and arrows indicate its direction. **k** Experimental mapping of local electrical resistance drops within the magnetoreceptor caused by the magnetic stylus nearby (first measured frame of the interaction shown in panel (**d**)).

magnetic fields with high resolution and low energy consumption. The simplified circuit design imparts a combination of optical transparency, mechanical compliance, and vapor/liquid permeability, rendering the magnetoreceptive mesh an imperceptible e-skin. Instead of focusing on each sensor readings at specific points, our magnetoreceptor captures electrical resistance information across the entire measurement domain, offering continuous spatial detection of magnetic stimuli across areas up to $120 \times 120$ mm$^2$ with a resolution below 1 mm. The scalability issues associated with the surging power consumption typically encountered with active-matrix systems[30,31] are largely mitigated, resulting in magnetoreceptive e-skins with nearly 500 times lower energy consumption. Our e-skins are promising for potential applications in virtual reality and augmented reality, enabling a new dimension for proximity, motion and orientation sensing (Fig. 1d). In combination with their immunity to external disturbance and ability for touchless interaction, our magnetoreceptors exhibit unique properties, as confirmed by the undisturbed recognition of fine-grained gesture through magnetoreceptive e-skin and the hygienic touchless interaction via a magnetoreceptive contact lens.

## Results

### Technical validation of the EMRT-enabled magnetoreceptor

Our large-area magnetoreceptor accomplishes the recognition of a magnetic stimulus by leveraging the electrical resistance tomography (ERT) in combination with the magnetoresistive medium, as summarized in Fig. 1e–g and Supplementary Figs. 1,2. Here, we coin this methodology as electrical magnetoresistive tomography (EMRT). Different from conventional point- or matrix-configurations where individual magnetic field sensors are distributed at specific points (Fig. 1a, b), EMRT employs a continuous giant magnetoresistive (GMR) multistack layer sputtered onto expansive flexible foils, the size of which varies based on the specific target area (Supplementary Figs. 2,3). Thanks to well-developed photolithography and sputtering techniques, the GMR film (which is strategically perforated for seamless integration onto biodynamic skins, as seen in Fig. 1h) is readily scaled up, for instance, to dimensions as large as $120 \times 120$ mm$^2$ in a laboratory setting, making it well-suited for practical application requiring operation over large areas. These magnetoresistive layers feature substantial electrical resistance reduction, owing to spin orientation when subjected to an external magnetic field (Fig. 1i)[36,37]. In principle, the spatial information of a magnetic source is discernible by extracting local changes in electrical resistivity within the magnetoreceptor during magnetic interactions (Fig. 1d). To implement this setup, a $70 \times 70$ mm$^2$ GMR mesh serves as the magnetoreceptive medium. The film is connected with 16 measurement electrodes, which are evenly distributed along its perimeter and facilitate indirect monitoring of local electrical resistance changes (Supplementary Figs. 1, 4). By applying a current to a specific set of electrodes, the resultant voltage, correlated to the electrical resistance information in the GMR film, is detected using another set of electrodes. By virtue of the substantial magnetoresistance response (approximately 23%) of the GMR layer, the injection current can be remarkably low (e.g., 1 mA), implying low energy consumption which is advantageous for wearable devices. Identical measurements are then iterated across the remaining sets of electrodes following a predefined measurement pattern (as depicted in Supplementary Fig. 4, Supplementary Table S1). In conjunction with finite element analysis, the voltage data stream is transformed into a map of discrete approximations (Fig. 1g). When utilizing a magnetic stylus (mechanical pencil filled with NdFeB permanent magnets) or similar input devices (e.g., magnetic skins consisting of polymeric elastomers and permanent magnet powders) for magnetic stimuli, the magnetic stray fields lead to localized drops in the resistance of the magnetoreceptor (Fig. 1j, k and Supplementary Figs. 5–8). The scanned voltage data is reconstructed into a continuous map,

exhibiting local resistance variations, through a resistive tomography reconstruction algorithm (please refer to methods)[38]. Rather than focusing on individual sensor readings at distributed points (Fig. 1a, b), the EMRT-enabled magnetoreceptor seamlessly captures the resistance data across the entire surface, enabling a holistic understanding of the resistance variation and the underlying magnetic interaction. The EMRT technique, which eschews discrete components and intricate connection wires within the sensing area, is anticipated to promote minimal power consumption, enhanced sensing resolution, while having a tidy and neat layout, resultant mechanical imperceptibility, visual transparency, vapor/liquid permeability for expansive magnetoreceptors, which will be thoroughly confirmed by the following research findings.

### High-resolution energy-efficient touchless interaction via EMRT-enabled magneto-receptor

Due to the ability of the EMRT-enabled magnetoreceptor to continuously sense magnetic fields across extended areas, it becomes feasible to propose gesture-based information communication without physical contact by interacting with a magnetic stimulus. In order to precisely locate a magnetic source, such as the magnetic stylus, the apex of the electrical resistance drop in the magnetoreceptor is correlated to the position of the stylus in terms of $x$ and $y$ spatial coordinates (Fig. 2a–c). This drop occurs due to strong magnetic fields concentrated around the tip of the stylus (Fig. 1j, Supplementary Fig. 5). The live magnitude of the resistance drop (Fig. 2d) and the temporal trajectory (Fig. 2e) of the apex are both captured and stored over time, which can be preprogrammed to trigger diverse complex commands. As a result, specific motion patterns such as graphical passwords for unlocking a smartphone can be precisely recognized and transmitted (Supplementary Movie 1 and Supplementary Fig. 9), as well as arbitrary motion of the magnetic stylus (Supplementary Fig. 10). In particular, handwritten characters such as "H" (Fig. 2c) and even composed whole words like "Hello" are successfully input by precisely tracking the $x$ and $y$ coordinates of the stylus hovering atop (Supplementary Fig. 11 and Supplementary Movie 2). To ensure reliable interaction, the EMRT-enabled magnetoreceptor requires a magnetic field threshold larger than 5–10 mT, which corresponds to a stylus-sensor distance of about 12 mm (Fig. 2f). This distance limitation can be extended by using a magnet with a more directed spatial profile of magnetic fields and/or using more sensitive magnetoreceptor based on the Py/Cu GMR effect (Supplementary Fig. 13) or tunneling magnetoresistance (TMR) effects[23,39], should applications require. Under the current working distance limit, the spatial position of the magnetic stylus can be reconstructed with a precision better than 1 mm (Fig. 2g, Supplementary Figs. 10,12). By integrating highly sensitive medium, spatially directed magnetic stimulus, and/or adjusting EMRT parameters (e.g., finite element size in analysis and number of measurement electrodes)[40], the sensing resolution can readily reach the sub-millimeter scale, and thus qualify for human motion recognition.

The attained spatial resolution (i.e., below 1 mm) over an elaborated area of $70 \times 70$ mm$^2$ for magnetic interactivity while retaining low power consumption presents a critical challenge for a conventional active matrix approach. From a circuit design standpoint, it necessitates the fabrication of a highly intricate array of multilayered devices encompassing $70 \times 70$ (m × n) sensor-transistor pairs, with an inter-element spacing no larger than 1 mm (Supplementary Note 1). Each sensing element will have to be addressed independently and requires power for operation, resulting in a quadratic escalation of the energy consumption with the sensing area when assuming unchanged device architecture and sensing pixel density. This hurdle significantly impedes the scalability of active matrices of magnetic sensors. In stark contrast, our EMRT approach maps magnetic fields using just one extended sensing element regardless of the scale of interactive area. The tomography algorithm, implemented into the EMRT approach,

performs a single scan of magnetic input (further referred to as frame) with only 208 sampling operations using 16 electrodes connected to the magnetoreceptor (Supplementary Fig. 4). As a result, to perform a single readout procedure, our magnetoreceptor consumes almost 500 times less power than the state-of-the-art matrix-based magnetic field sensors[30,31], as indicated by a comparison using standard operating parameters (Supplementary Note 1, Supplementary Table S2). Hence, a 9-fold increase in EMRT-driven magnetoreceptor area results in only 20% increase in energy consumption (Fig. 2h). The low power budget (and consequently reduced heat accumulation) is experimentally confirmed by negligible temperature fluctuation in our magnetoreceptor even after continuous operation for tens of minutes (Fig. 2i).

## Visual transparency, mechanical compliance, and vapor/liquid permeability of EMRT-driven magnetoreceptor for imperceptible e-skin

Large-area electronic skins intended for seamless integration onto human skin should not compromise other essential skin functions

such as reflecting visual health information, accommodating mechanical movements (e.g., bending, stretching), and facilitating vapor/liquid exchange with environments. However, conventional active-matrix sensing requires sophisticated multilayered electrical circuits and a substantial number of external electrodes. While our EMRT skin utilizes only 16 measurement electrodes (with the possibility of downscaling to 8 electrodes, see Supplementary Movie 3), a minimum of 140 external electrodes would be required for active matrix sensing (Supplementary Fig. 14, Supplementary Table S2)[30,31] with a sensing resolution comparable to our magnetoreceptor. These wires not only act as antennas that induce electromagnetic noise but also impair the mechanical deformability, visual transparency, and vapor/liquid permeability, therefore impeding the imperceptible integration of the magnetoreceptor as e-skin. The tidy and organized layout facilitated by the EMRT-enabled magnetoreceptor effectively addresses these issues simultaneously.

The GMR film is lithographically patterned into micrometric meshes and meticulously fine-tuned in its geometrical parameters in

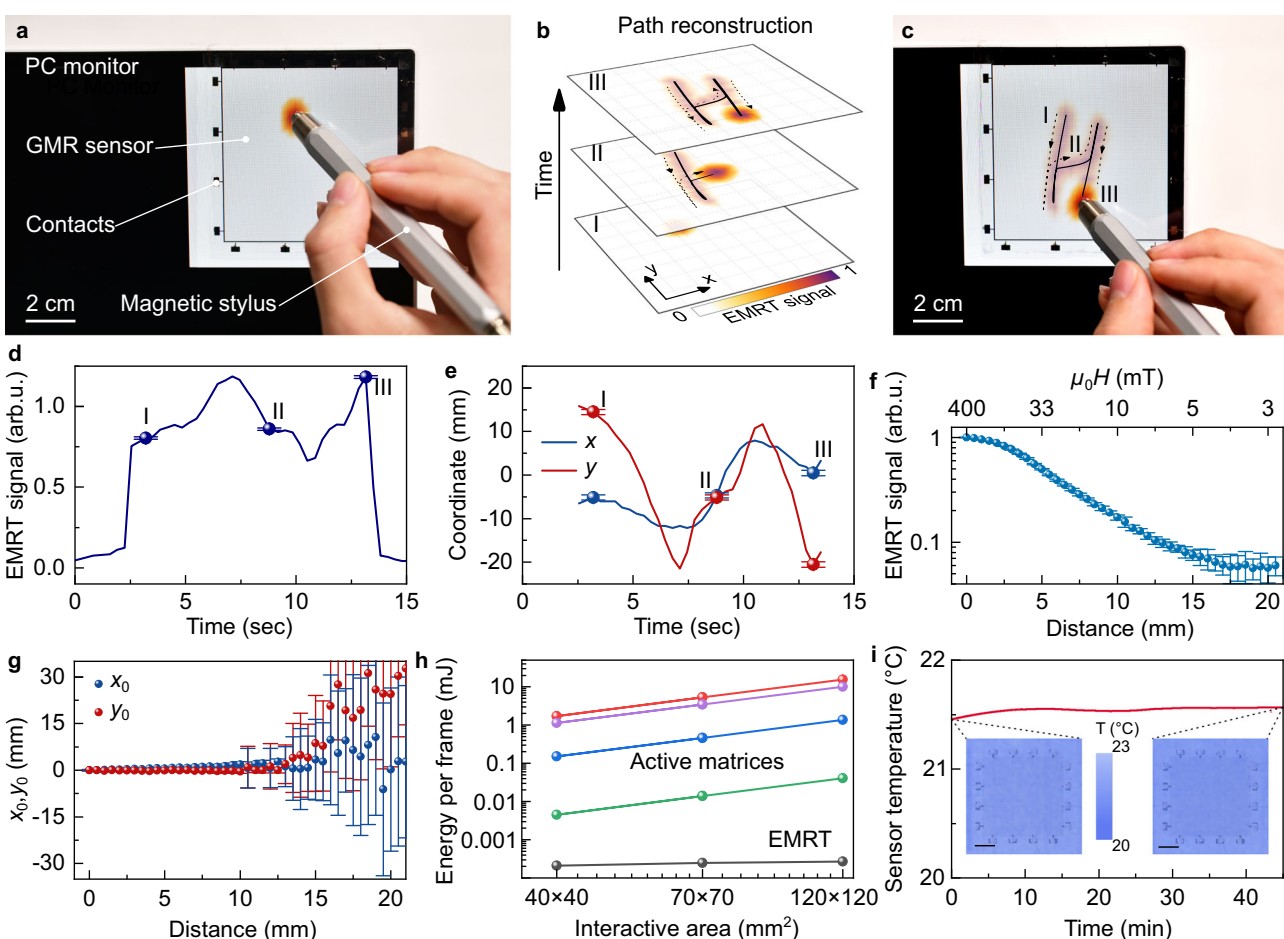

**Fig. 2 | Touchless interaction based on EMRT-enabled magnetoreceptor. a** A photograph of the transparent magnetoreceptor applied to the computer monitor and used for human-machine interaction, in this case handwriting input of a word "Hello" (see also Supplementary Movie 2). **b** A set of reconstructed frames from the input of letter "H" depicted in Supplementary Movie 2. **c** A photograph of resulting letter "H" handwritten using EMRT-enabled magnetoreceptor. **d** Variation of the EMRT signal amplitude in an interaction with a magnetic source. **e** x, y plot of the reconstructed trajectory of a magnetic source moving over the magnetoreceptor while handwriting the letter "H". The solid lines in panels (**d**) and (**e**) depict the full recorded data set and symbols indicate values for the selected frames in the panel (**b**). **f** The magnitude of EMRT signal with increasing the distance between the magnetic stylus tip and the magnetoreceptor plane. **g** Variation of reconstructed position of the magnetic stylus ($x_0$ and $y_0$ coordinates) with increasing distance

between the magnetic stylus tip and the magnetoreceptor plane. The symbols and error bars in the panels (**f**) and (**g**) denote the mean value and standard deviation of the EMRT signal amplitude and reconstructed (x,y) position over 30 recorded frames. At distances above 15 mm the EMRT algorithm is not able to reliably localize magnetic input due to weak GMR response (below some percent). **h** Comparison of energy consumption scaling with increasing interactive area between different active transistor matrices and our EMRT platform. The energy consumption of the active matrix-based devices is estimated relying on the reported operating parameters of individual sensing element provided in reports: red[30], blue[53], green[54] and purple[55] (Supplementary Note 1). **i** Temperature of the magnetoreceptor powered with 2 mA current. Inset images are taken with a thermal infrared camera in the beginning and the end of the measurement. The colorcode corresponds to the temperature distribution over the sample area.

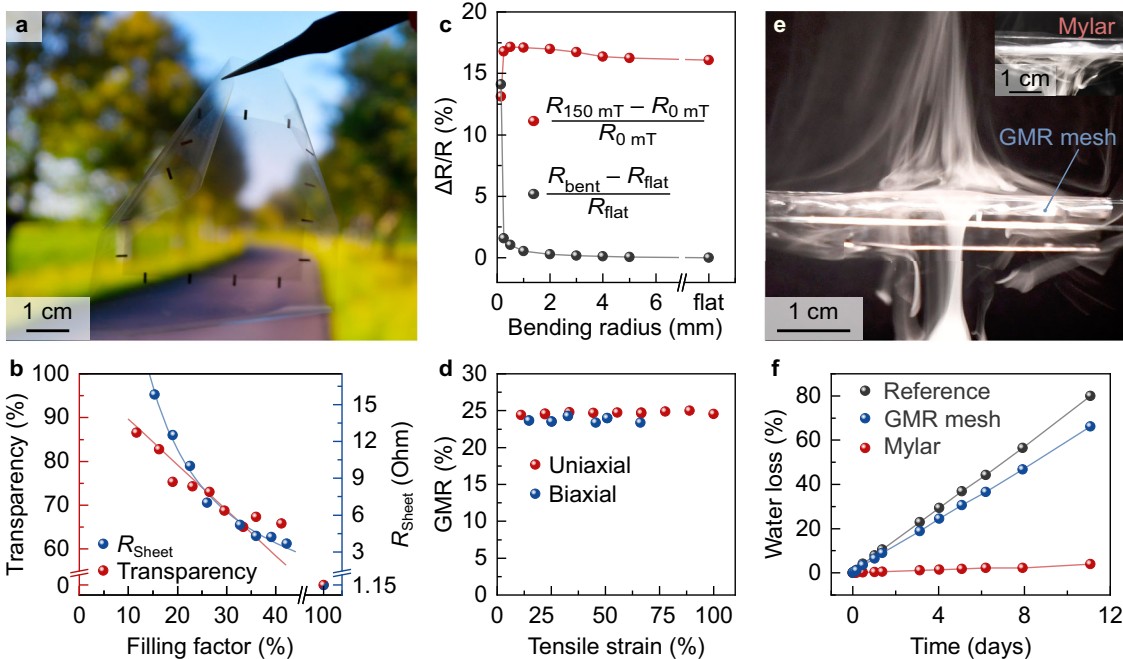

**Fig. 3 | Imperceptible magnetoreceptive e-skin. a** Optical photograph of a transparent flexible magnetoreceptive mesh. **b** Light transmission and sheet resistance of GMR meshes as a function of filling factor. **c** GMR ratio and electrical resistance of the GMR mesh as a function of bending radius. **d** Evolution of the GMR response with application of uni- and biaxial mechanical strain to the GMR mesh. **e** Visualization of the enhanced permeability of the GMR mesh. Inset: permeability of a standard polymer foil. The vapor flow patterns show the effect of the breathable mesh. **f** Vapor permeability of the mesh measured by water evaporation and compared with a reference (no barrier) and a standard non-permeable foil.

order to achieve optical transparency while preserving optimal magnetoreceptive performance. A pitch of 100 μm as used for all GMR meshes ensures high sensing resolution. Changing the filling factor $f$ (via linewidth optimization) permits the adjustment of the optical, electrical, and magnetoresistive properties of the e-skin. The light transmission of the magnetoreceptive mesh can be improved by lowering the filling factor at the cost of ~ $1/f$ increase in resistivity (Fig. 3a, b, Supplementary Fig. 15). We chose a GMR mesh filling factor of about 20 % to achieve a balanced compromise with light transmission of about 75% alongside a sheet resistance of about 12 Ohm. The selected 100 μm pitch in combination with 10 μm linewidth surpass the visual acuity threshold for human vision and thereby ensures the undisturbed optical transparency of the e-skin (Fig. 3a). Importantly, altering the filling factor has no impact on the magnetoresistance of the GMR mesh (Fig. 3b), allowing for optical transparency without compromising magnetoreceptive performance. This unique blend guarantees the unimpaired functionality of the magnetoreceptor in various situations, including on transparent objects, light-emitting surfaces, or human tissues.

The monolayer configuration and scalable circuit design inherent to our EMRT technique ease integration onto ultra-flexible substrate materials (here sputtered on a 3 μm thick Mylar foil) and thus endow the magnetoreceptor with exceptional, yet reliable mechanical deformability. Mechanical assessments confirm excellent robustness, revealing stable electrical resistance and magnetoresistance even when the magnetoreceptor is bent to a radius of curvature as small as 250 μm (Fig. 3c, Supplementary Fig. 16). Furthermore, when adhered atop a pre-stretched elastic membrane, our magnetoreceptor can withstand up to 100 % uniaxial and up to 75% biaxial elongation (Fig. 3d, Supplementary Figs. 17,18) due to the formation of an intricate network of wrinkle patterns that enable geometric stretchability[41,42]. This combination of flexibility and stretchability endows our magnetoreceptors with the mechanical conformability required for seamless integration onto human skin for wearable magnetoelectronic applications.

In addition, our magnetoreceptor gains fluid and vapor permeability owing to the meshed layout and the low areal filling factor of functional GMR materials that introduce periodic openings. Here, oxygen plasma is applied to selectively etch the supporting polymer in areas devoid of GMR metals that in turn act as etch mask to create an array of holes. This configuration amounts to roughly 10,000 holes/cm², approximately 20 times denser than human skin pores[43]. Due to a low filling factor of about 20%, the magnetoreceptor allows moisture, vapor, and sweat to pass through quite easily (Fig. 3e, Supplementary Movie 4). A magnetoreceptive mesh of this configuration is almost as permeable as having no barrier, and outperforms both polymer foils and a regular medical plaster (Fig. 3f, Supplementary Figs. 19,20). This enhanced permeability has the potential to improve wearing comfort and reduces the risk of skin irritation or other dermatological issues, especially during prolonged use of on-skin magnetoreceptors[44,45].

### Undistorted perception of fine motion in extended reality via magnetoreceptive e-skin

Due to the intricate combination of mechanical compliance, vapor/liquid permeability, low energy consumption, good sensing resolution our magnetoreceptors can be comfortably worn on-skin to serve as interfaces for the precise transmission of gestural information with various electronic devices over extended periods of time. One possible use case that would benefit from the implementation of our magnetoreceptive e-skins is Extended Reality (XR). XR is an umbrella term for Augmented Reality (AR) and Virtual Reality (VR) digital worlds that allow users to have immersive experiences. Immersion is often achieved through visual rendering and responsive sensors that embed the user in the XR. For example, away from physical controllers, many consumer VR products rely on optical sensors. The limited field of view and low spatial resolution of optical sensors render capturing fine-grained motion and providing accurate feedback difficult. This can be resolved by incorporating high-resolution optical sensors, however at an often prohibitively high cost. The spatial continuous sensing capabilities of the EMRT-enabled magnetoreceptors aim to address this

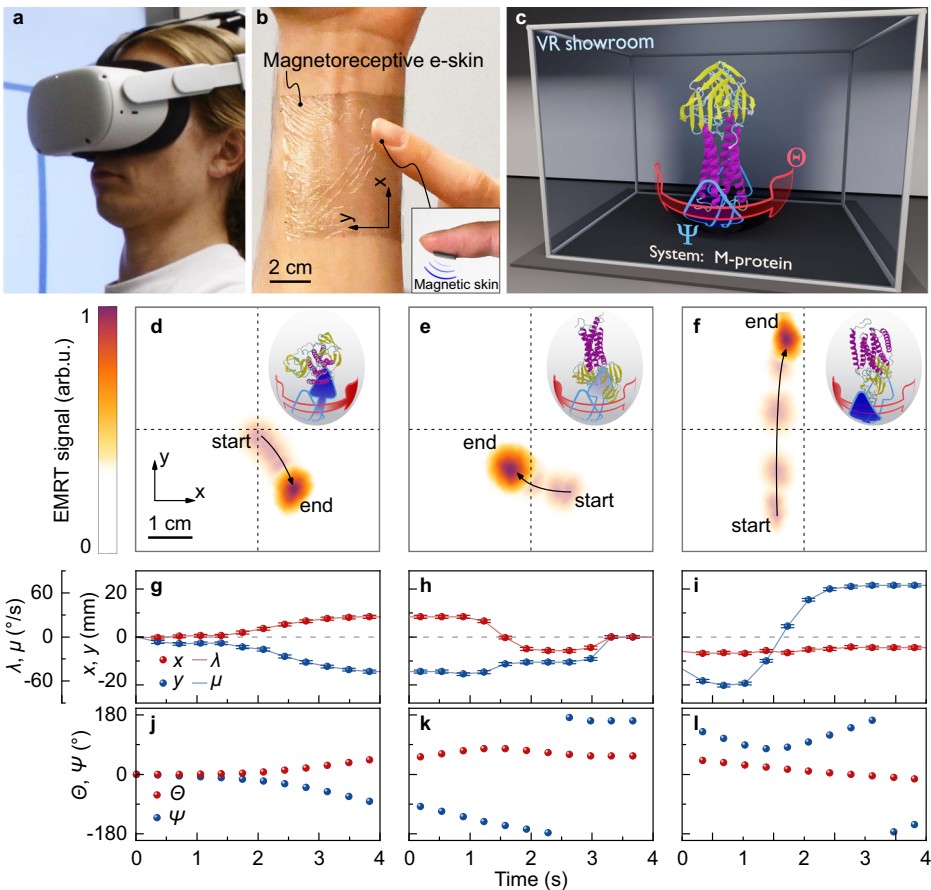

**Fig. 4 | Magnetoreceptive mesh for high-precision virtual reality (VR) applications. a** Photograph of a person wearing a VR headset. **b** Mechanically permeable imperceptible magnetoreceptive e-skin applied to a wrist of a person. Inset: magnetic skin made of NdFeB composite applied to the fingertip. **c** VR showroom experienced by a person in panel (**a**). **d**–**f** Selected data from the interaction shown in Supplementary Movie 5. The semi-transparent trace is the trajectory of the magnetic input and the opaque spot stands for the last frame of the corresponding segment. The insets in the upper right corners highlight the direction of rotation at last frame of the segments (See methods for more details). **g**–**l** Timeline of the EMRT reconstructed magnetic input coordinates $x$ and $y$ (symbols) and recalculated angular velocities λ and μ (lines), corresponding to azimuthal and polar rotation angles $\Theta$ and $\Psi$. The error bars in panels (**g**), (**h**) and (**i**) depict the uncertainty of the reconstructed magnet position of 0.6 mm, obtained from the data shown in Fig. 2.

challenge by detecting subtle movements in the vicinity of functional areas and accurately transmitting them to the virtual environment.

In Fig. 4a, a person wearing a standalone VR headset and our magnetoreceptive e-skin (Fig. 4b) experiences an educational VR showroom depicted in Fig. 4c. The task at hand is to rotate the molecule in the molecularium (molecular aquarium). A large-area magnetoreceptor (measuring $55 \times 55$ mm² in lateral dimensions), seamlessly affixed onto the user's arm, is utilized to control the orientation of a virtual object (here, a protein molecule) in three-dimensional space (Fig. 4d–l, Supplementary Movie 5). A small magnetic skin made of NdFeB composite is also attached to the user's finger to enable magnetic interaction (Fig. 4b). Since the EMRT-driven e-skin detects the magnetic stimuli by means of $x$ and $y$ spatial coordinates (in two dimensions), the virtual object exhibits two degrees of freedom denoted as $\Theta$ and $\Psi$, corresponding to azimuthal and polar rotation in two orthogonal planes (Fig. 4c, with Fig. 4d–f showing specific interaction segments selected from the Supplementary Movie 5). Figure 4g–i display the time variation of the reconstructed magnetic input coordinates $x$ and $y$ and the logic behind the orientation manipulation of the virtual protein molecule. The spatial position of the magnetic trigger is localized relative to the center of on-skin magnetoreceptor (which signifies the position $x = y = 0$). The $x$ and $y$ positions are then recalculated into angular velocities λ and μ, that correspond to azimuthal ($\Theta$) and polar ($\Psi$) rotation (Fig. 4c). Figure 4j–l shows the resulting time variation of the $\Theta$ and $\Psi$ angular

positions of the protein molecule depicted in the Supplementary Movie 5. Such sophisticated scenarios typically involve gestures where the hands of the user are crossing, which prohibitively complicates fine-motion reconstruction of the fingers from optical tracking alone. The incorporation of our EMRT-enabled magnetoreceptor successfully improves the sensing in such complex environments, enabling smoother and more accurate detection of fine-grained motion tracking in VR. Notably, our magnetoreceptor allows for accurate reading and localization of multi-channel magnetoelectrical signals when magnetic stimuli are applied at two or more different positions simultaneously (please refer to Supplementary Figs. 7,8 and Supplementary Movies 6,7). The multi-channel reading capability facilitates the development of more advanced interactive devices. This technical advance facilitates the creation of more immersive XR experiences. Compared with traditional interactive methods (e.g., through optical, touch, capacitive sensing), our magnetoreceptors offer unique advantages in XR, i.e., remaining functionally unaffected due to magnetic immunity to disruptions caused by sweat, moisture, or opaque objects[46–48] (see Supplementary Figs. 21, 22 summarising the sensor performance at different environmental conditions). Such extensive adaptability allows operation not only on bare skin for sensing and tracking, but also through garment or other barriers, even in swimming pools or on rainy days, that typically hinder traditional on-skin sensing systems, providing users with uninterrupted and versatile ways of interaction with digital worlds.

The large-area magnetosensitive e-skin presented in our manuscript is designed to work in a *contactless* regime relying on the interaction with magnetic stray fields. Therefore, the device is designed not to provide pressure readings. The insensitivity of the magnetoreceptive GMR mesh to the pressure and mechanical deformation is demonstrated in Supplementary Movie 6, where the sensor responds only when it is approached by fingers fashioned with magnetic skin and does not show any response when pressed by bare fingers. Additionally, the mechanical stability of the GMR mesh upon mechanical stimuli is shown by bending and stretching experiments shown in the Supplementary Figs. 16–18. We note that electrical impedance tomography is already widely used to realize touch and pressure sensors[38,40]. Our touchless on-skin sensor technology may be readily combined with these pressure sensitive devices to realize a multifunctional sensor with touchless and tactile sensitivity[22].

## Touchless magnetoreceptive contact lens for hygienic interaction in augmented reality

Enabled by features such as optical transparency and touchless interaction, our magnetoreceptors are capable of fostering prospective applications. We demonstrate these possibilities by introducing a magnetoreceptive contact lens. Here, a flexible and transparent magnetoreceptive mesh is attached to a contact lens, where the transparency of the mesh does not impair the original optical functionality, but the ability to reconstruct complex spatial patterns is added in order to serve as an input device for interacting with augmented reality scenarios (Fig. 5a, b, Supplementary Movie 8). Importantly, the magnetic field enabled touchless operational mode of such magnetoreceptive interfaces may help in maintaining eye hygiene and reduce the risk of infection spreading during intense and intimate interaction. This concept is illustrated by augmenting an actual image with a selection marker, to pick specific points of interest in a touchless manner (Fig. 5c–e). The marker becomes active once the user approaches with a magnetic object (e.g., stylus) and crosses the first proximity threshold, entering the "Navigation" mode (Fig. 5e). At this distance from the mesh, the user can freely move and select the region of interest (e.g., upper part of the building in Fig. 5c), and then confirm it by approaching closer to the mesh, thus activating the "Selection" snapshot function (Fig. 5e). The taken snapshot can then be digitally zoomed in and out by moving a magnetic stylus along the perimeter of the contact lens and mapping its trajectory (Fig. 5f–h). Specific positions of the stylus were recognized by software and interpreted to magnify the image by factors of 1×, 2.5× and 5×, respectively (Fig. 5i and Supplementary Movie 8). Key to this zooming process is the ability of the magnetoreceptive mesh to distinguish the angular (positional) information of the stylus as it remains within the proximity threshold corresponding to the snapshot mode. Our EMRT-enabled magnetoreceptor align well with the increasing emphasis on non-contact interactions, promoting a safer environment in XR, smart medicals, and IoT in general.

We note that the magnetic pen can be replaced with magnetic skins made of mechanically soft magnetic composites (Supplementary Fig. 7). These magnetic skins can be easily attached to fingers, serving as a source of magnetic stimuli and reducing the impact on the field of view caused by the magnetic pen during operation. Using magnetic skins on a finger has an important advantage: Since two eyes provide stereo vision, partial obstruction of the view of one of the eyes should not cause significant discomfort during interaction with the device. Next steps should also include demonstration of wireless communication with the magnetoreceptive contact lenses[49–51].

## Discussion

This work develops an approach for spatially continuous high-resolution magnetic field mapping by employing the EMRT technique based on an extended magnetoreceptive medium and electrical

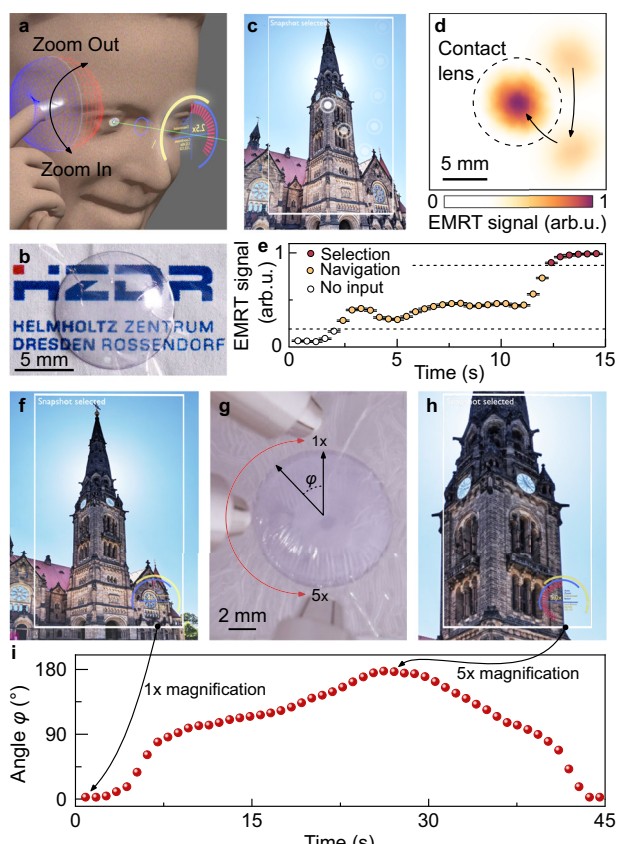

**Fig. 5 | Magnetoreceptive contact lens for augmented reality applications.**
**a** Concept image showing how magnetically responsive interaction layers could be used for, e.g., zooming in or out in augmented (virtual) environments. **b** Optical micrograph depicting the transparency and mechanical compliance of the magnetoreceptive mesh applied on a contact lens. **c–e** Selection of a snapshot image using magnetoreceptive contact lens. Selection marker is moved over the field of view (c) using spatial reconstruction of the magnetic trigger over the lens area (**d**). **e** EMRT signal range is divided into three sections: no input detected (white symbols); marker selected (orange symbols); snapshot selected (red symbols). The error bars depict the uncertainty of the EMRT signal magnitude based on the data shown in Fig. 2. **f–i** Digital magnification of the selected snapshot. **g** Zoom-in motion of a magnetic stylus around the perimeter of the magnetoreceptive contact lens. **f, h** Representative frames of Supplementary Movie 8 revealing a sequentially zoomed-in picture based on the interaction depicted in (**g**). **i** Corresponding dependency of reconstructed angular position φ of the magnetic stimulus.

resistance tomography. Unlike conventional active-matrix configurations with their constraints of distributed sensors and assistive transistors, this approach does not require active transistor matrices and complex sensor arrangements. EMRT provides more freedom for users to interact with and "write" over the sensing surface without increase in circuit complexity, power consumption and fabrication cost. Consequently, our technology unlocks magnetic smart e-skins as an additional perception tool for a broad range of devices and objects, including on-skins XR interfaces, light-emitting surfaces and contact lenses. The combination of fine spatial resolution, lightness, low power consumption, undisturbed function, and contactless operation renders our magnetoreceptive e-skin an appealing constituent of future XR controllers beyond capacitive/pressure based sensors. Further, the robust touchless interaction enabled by our magnetoreceptor can effectively extend the perceptual range for people with disabilities. For instance, blind individuals equipped with the magnetoreceptor system could extend the distance of their perception, and those with prosthetic hands could use it to interact with smartphones, overcoming the challenge of insulated prosthetics not being able to interact with

capacitive touchscreens. In a more distant future, in conjunction with advanced neural interfaces, magnetoreceptive e-skins may help to endow humans with a perception to magnetic fields and enable full-fledged magnetoception.

## Methods

### Preparation of transparent and water-permeable GMR sensor films

GMR sensors were conceived as meshes with a tunable linewidth (6–25 µm) and a fixed pitch of 100 µm. The mesh is patterned over areas up to $120 \times 120$ mm$^2$, providing equally distributed contacts along their perimeter (Supplementary Fig. 2b). 2.5 µm- and 125 µm-thick polyester foils (Mylar, DuPont USA) were used as substrates to fabricate transparent GMR sensors based on standard photolithography. Mylar films were affixed onto polydimethylsiloxane (PDMS) coated glass plates, so they could be processed in a flat state without crumpling. The photolithographic process consisted of: oxygen plasma for 30 s, spin-coating at 4000 rpm with AZ5214E photoresist (MicroChemicals GmbH, Germany), soft baking for 5 min at 90 °C, exposure with a direct laser writer (DWL 66, Heidelberg Instruments, Germany), post-exposure bake for 2 min at 120 °C, flood exposure and development in 1:4 solution of AZ351B developer (MicroChemicals GmbH, Germany) in deionized water for 60 s. Multilayered stacks of [Co (1 nm) /Cu (2.2 nm)]$_{50}$ coupled at the 2nd antiferromagnetic maximum were deposited on foils by dc magnetron sputtering at room temperature (Ar was used as a sputter gas at a pressure of $8 \cdot 10^{-4}$ mbar; base pressure: $10^{-7}$ mbar; deposition rate: 2 Å/s). After deposition, the samples were lifted off in acetone and cleaned with isopropanol to reveal the desired mesh patterns. Selective etching of Mylar foil was performed using magnetosensitive element covered with additional layer of Ta (5 nm) as a shadow mask, by subjecting the samples to oxygen plasma (Femto plasma cleaner, Diener Electronic, Germany): the generator power was set to 400 W at pressure of 0.3 mbar. Etching was done for 45 min.

Long-term safe and stable operation of e-skins is ensured by the use of appropriate encapsulation layers that are suitable for the targeted use case scenario. In the case of magnetoreceptive skins, potential hazards are related to the use of cobalt-based magnetic sensing layers and circulating electric currents in resistive sensors. Hence, the use of insulating encapsulation later is imperative for our sensors when aiming on-skin applications. For encapsulation, we used plastic varnish (Plastik70, CRC Industries UK Ltd. United Kingdom). However, other encapsulation layers, which are typically used in flexible on-skin electronics like PDMS and ecoflex can be readily applied as well. In particular, the encapsulated sensors operate in a broad temperature range from 10 to 60 °C (typical operation temperature range for on-skin devices; Supplementary Fig. 21) and can be submerged into liquids with different pH (mimicking exposure to physiological liquids, like sweat, etc.; Supplementary Fig. 22).

### Transparency measurements

To quantify the dependence of the optical transparency on the linewidth of the sensor mesh, a series of $1 \times 1$ cm$^2$ meshes with different linewidth were prepared on PET foils (Supplementary Fig. 15). A reflective neutral density optical filter (Thorlabs, USA) was placed between a light source (HXP 120 V, Zeiss, Germany) and the sample, to reduce the intensity of the source. The incoming light was transmitted through the sensor films, collected and analyzed with a spectrometer (USB 650, Ocean Insight, USA). Two segments of optical fiber were placed before and after the sample to guide and gather the incoming and transmitted light, respectively. Transparency values were calculated as a ratio of integral intensities within visible electromagnetic spectra (380–680 nm) transmitted through the mesh and blank PET foil.

### Magnetoresistive characterization

The magnetic responses of the transparent GMR sensor mesh was characterized by applying an in-plane external magnetic field $H_{ext}$ using an electromagnet, powered by a bipolar power supply (Kepco, USA). The longitudinal resistance of the sensor mesh was measured in a 4-wire configuration, using a Tensormeter (HZDR Innovation, Germany). Frequency and amplitude of the driving current were 775 Hz and 100 µA, respectively. The GMR ratio was defined as the magnetic field dependent change of the sample's resistance, $R(H_{ext})$, normalized to the value of resistance in magnetically saturated sample, $R_{sat}$: $GMR(H_{ext}) = [R(H_{ext}) - R_{sat}] / R_{sat}$.

### Mechanical characterization

The mechanical stability of the sensors was characterized upon static bending and stretching tests:

a. Static bending test: Transparent sensors prepared on 2.5 µm-thick mylar foils were mounted on cylindrically shaped sample holders (Supplementary Fig. 16) with curvature radii ranging from 250 µm to 5 mm and placed in between pole shoes of an electromagnet. To ensure uniform field in the sensing plane, the sensors were mounted with their curvature axes perpendicular to the axis of pole shoes. The magnetic field of the electromagnet was swept between −200 and 200 mT and the GMR response of the sensors was simultaneously recorded.

b. Uniaxial stretching test: GMR sensor meshes were affixed onto viscoelastic tape (VHB, 1-mm-thick, 3 M, USA), which was pre-stretched to 500% with an initial length of 20 mm. The longitudinal edges of the tape were attached to two movable clamps, used to tense and release the samples. The elastic GMR sensor was released until the longitudinal dimension of the sensor mesh was 10 mm. This state was taken as initial for strain calculation. During the experiments, the elastic GMR sensor was stretched to different values of strain up to 100%. For each strain value, its GMR characteristic was measured as for the case of static bending tests.

c. Biaxial stretching test: Elastic sensors were adhered to a VHB tape pre-stretched to a diameter of 150 mm (final thickness of about 0.5 mm, initial diameter: 50 mm). The sensor was affixed at its edge using 8 radially distributed clamps (Supplementary Fig. 18). By synchronously moving all clamps, a biaxial strain was applied to the sample. Simultaneously, the change in a 4-wire resistance of the sample was measured using a Digital multimeter (34461 A Truevolt, Keysight Technologies, USA). To assess the sample magnetoresistance, an external magnetic field of about 200 mT was applied to the sensor using a permanent magnet. This field is sufficiently strong to magnetically saturate the sensor and access the maximum GMR value. The GMR ratio was defined as a difference between the initial resistance ($R_{init}$) and with approached magnet ($R_{magn}$), normalized to the $R_{magn}$: ($R_{init} - R_{magn}$) / $R_{magn}$.

### Electrical magnetoresistive tomography (EMRT)

A square-shaped large-area transparent GMR magnetoreceptor is electrically connected along its periphery with 16 evenly distributed electrodes (4 electrodes per side). The number of electrodes and their positioning can vary according to specific requirements and magnetoreceptor geometry. The electrical sampling is done following the algorithm: an electrical current is sequentially injected through the pairs of adjacent electrodes and the resulting voltages are measured between other pairs of neighboring electrodes (Supplementary Fig. 4). The sequence is repeated until all the electrode pairs are probed and the corresponding voltage dataset is collected (Supplementary Table S1). This process is driven by a custom-built switching matrix module composed of PXI 2535 matrix switch and PXI3-4303 analog input modules (National Instruments, USA) (Supplementary Fig. 1)

controlled by a LabVIEW (18.0f2 (64-bit)) program. Next, the measured voltage dataset is used to reconstruct the electrical resistance (conductivity) map. This is done using the EIDORS (Electrical Impedance Tomography and Diffuse Optical Tomography Reconstruction Software)[52], an open and free software algorithm for forward and inverse modeling for Electrical Impedance Tomography driven by Matlab (R2018b (64-bit)). The reconstruction is performed in the differential mode, as this approach allows to isolate the GMR-induced resistance change from unwanted external influences (such as contact resistance variations). For this, the first measured dataset is used to account for the baseline of the resistivity map and the subsequent measurements show the local GMR response induced by a magnetic trigger, e.g. magnetic stylus or magnetic skin. During the reconstruction, a 2D finite element model (FEM) of the magnetoreceptor is built (Fig. 1g) and the resistivity map is obtained using an iterative Gauss-Newton algorithm[52]. The obtained resistivity map is then post-treated according to the pre-defined use-case scenario and visualized using a Wolfram Mathematica script. If only one magnetic input is anticipated, the position of the magnetic stimulus is attributed to the FEM element with the highest conductivity change. A more advanced analysis is required to recognize multiple simultaneous magnetic inputs. In this work, it is realized using a multi-Gauss fit of the reconstructed resistivity map. Spatial positions of each peak are interpret as interaction point (Supplementary Fig. 7 and Supplementary Movie 6).

### Seamless body sensor for immersive VR

To illustrate another potential use of the GMR body-based interactive elements in AR and VR experiences, we have designed a simple application utilizing the Unity3D platform. In this example, by using the OpenVR plugin for Unity's XR API and standard android SDK, we have created a showroom where a user can transfer a 3D object of interest in the manipulation area and use the seamless body sensor as a physical guide to manipulated polar angles of the said object. The communication between the sensor elements and VR headset was realized through the 5 GHz Wi-Fi network.

### Magnetically-enabled touchless writing using EMRT

A sensor mesh was placed face down at a corner of a monitor (Supplementary Fig. 11) to ensure no physical contact between the magnetic stylus and the mesh. The sensor was connected to a custom-made switch matrix based on three PXI modules by National Instruments (4322 as analog output, 2535 for switching, 4303 as analog input) to automatize ERT measurements in areas with up to $7 \times 7$ cm² and 16 electrical contacts (Supplementary Fig. 1). A clutch pencil was turned into a magnetic stylus by replacing the graphite lead with a NdFeB spherical magnet (HKCM engineering, Magnet-Sphere K05Cr-N45, Article 9960-425) of 5 mm in diameter and generating a magnetic field of about 350–400 mT on its surface, as measured with a gaussmeter (HGM09s MAGSYS, Germany). During touchless writing, the magnetic stylus hovered over the sensing area at a distance in the range of 5 mm. The trajectory of the magnetic stylus was digitized by extracting the maximum drop of the electrical resistance at each taken resistance map and drawing a point on the screen at that position. Successive points were connected with a line in Wolfram Mathematica Graphics Window to depict characters and letters on-screen.

### Magnetoreceptive contact lenses

Commercial contact lenses (10 mm in diameter, Volens, Germany) were used as support and frame for the experiments. By optimizing the degree of pre-strain applied to the viscoelastic tape, the elastic and transparent GMR sensor was conformably attached to the lens (Fig. 5b). Connections to the sensor matrix were accomplished using 50 μm-thick copper wires. A smaller cylindrical magnet

(HKCM engineering, Magnet-Cylinder Z02x02Ag-N35, Article 9962-316) was attached to the magnetic stylus to facilitate manipulation over a more size-restricted area. The electrical outputs of the transparent GMR mesh were read using the same ERT hardware and software as described above for the writing experiments. The software was expanded to include a zooming interface by assigning a sequence of images with different magnification to particular azimuthal angles of the magnet position, with respect to the center of the lens. The zooming range was defined as the span between magnifications of 1× and 5×, corresponding to opposite positions of the magnetic stylus on the lens (Fig. 5i). As a proof of concept, an image was zoomed in and out using this transparent contact lens interface.

### Evaluation of water vapor permeability

Water vapor permeability was evaluated by measuring water loss from the glass flask bottleneck of which was covered with target material. Pure water (2 g) was placed into a flask with bottleneck diameter of 10 mm. The GMR mesh was fixed onto flask bottleneck with Parafilm "M" (Bemis Company Inc., USA). The flask was stored under a fume hood on a hot plate at 35 °C to simulate the approximate temperature of the human eye. A medical plaster (aluderm®-aluplast, elastic, aluminized, W. Söhngen GmbH, Germany) and continuous PET film were used as reference samples. To visualize the permeability of the magnetoreceptive mesh we affixed the mesh on top of a container, in which we placed cotton wool soaked with a solution of propylene glycol and glycerin, which was burnt with a heating coil. The upward flow of the steam coming from the wool was visualized in a dark background and the patterns for the mesh and a standard polyester foil (Mylar, DuPont USA, 2.5 μm-thick) were compared (Supplementary Fig. 19h, i). We performed grazing angle imaging in order to show that our permeable GMR mesh sensor does not obstruct the vapor flow. For this experiment, a partially perforated GMR mesh sensor was prepared. Drop Shape Analyzer – DSA25E (KRÜSS GmbH, Germany) setup is used to visualize the surface of the sample in the 10° grazing angle geometry. During the experiment warm water (about 60 °C) was poured into the container located below the sensor. On the perforated side of the sensor vapor flow easily penetrates the perforated mesh, no condensate is formed on the sensor surface. Conversely, condensation quickly forms in the non-etched part (Supplementary Fig. 20 and Supplementary Movie 4).

### Joule heating of the magnetoreceptor

Heating effects during sensor operation were monitored using thermal camera (TrueIR Thermal Imager U5850 Series, Keysight Technologies, USA). Two set of experiments were performed: (i) a sensor mesh (10 μm linewidth, 100 μm pitch) of $2 \times 2$ cm² size was connected to a custom-made switch matrix device running at constant current of 2 mA for 45 min. (ii) A $1 \times 1$ cm² size GMR mesh samples with different linewidth (used for transparency measurements) were biased with gradually increased current by 1 mA every minute from 1 to 20 mA and temperature was continuously monitored (Fig. 2i).

### Ethics

We applied a permanent magnet to a finger of a user in several studies reported in this manuscript. This magnet is not an electronic component. Furthermore, in several studies, a sensor is applied on skin of a user. These studies are done accordingly to the ethic approval #SR-EK-459122024 from the ethics committee at the Technical University of Dresden. For this case, we have a written consent of the user (one volunteer, male, 27 years old), which was wearing this sensor. The current lines are always isolated and are not in touch with the skin. The sensor is not worn on skin for any extended duration. The authors affirm that human research participants provided informed consent for publication of the images in Fig. 4a.

**Reporting summary**

Further information on research design is available in the Nature Portfolio Reporting Summary linked to this article.

## Data availability

All of the data supporting the conclusions are available within the article and the Supplementary Information. Additional data are available from the corresponding authors upon request. Source data are provided with this paper.

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

## Acknowledgements

We thank Rainer Kaltofen, Conrad Schubert and Dr. Jens Ingolf Mönch (all HZDR) for the support with the deposition of metal layer stacks. We acknowledge the input and discussions by Dr. Tobias Kosub (HZDR), Prof. Minjeong Ha (HZDR) and Dr. Uwe Vogel (Fraunhofer Institute FEP) during the development of this project. We express our gratitude to the Research Technology department (Nicole Wagner and Bert Lange) at the HZDR for the support with the development of the EMRT hardware. Support by the Structural Characterization Facilities Rossendorf at the Ion Beam Center (IBC) at the HZDR is greatly appreciated. This work is financed in part via the German Research Foundation (DFG) Grants MA 5144/13–1 (D.M.), MA 5144/28–1 (D.M.), Helmholtz Association of German Research Centers in the frame of the Helmholtz Innovation Lab "Flexi-iSens" (D.M.), European Commission HORIZON RIA (project REGO; ID: 101070066; D.M.), and ERC grant 3DmultiFerro (Project number: 101141331; D.M.).

## Author contributions

J.G. conceived the idea of transparent GMR sensors. P.M. and J.G. fabricated and characterized transparent GMR sensors. P.M., G.S.C.B, R.I. R.X. and D.M. analyzed the data and prepared figures with contributions from all authors. L.I., D.M., Y.Z. and P.M. put forth the concept and realized breathable magnetic field sensors. D.M., G.S.C.B., Y.Z., and P.M. designed and realized the concept of magnetoreceptive contact lens. O.V. wrote the processing algorithms for EMRT and contributed with the analysis of the EMRT data. J.G. developed the measurement setup for EMRT. P.M., J.G., Y.Z., M.K. and D.M. designed and prepared demonstrators using transparent sensors. S.A. developed the virtual reality demonstrator. R.X., P.M., G.S.C.B, M.K. and D.M. wrote the manuscript with comments from all authors. All co-authors edited the manuscript. D.M., L.I., M.K. and J.F. conceived the project.

## Funding

## Competing interests

The authors declare no competing interests.
