## [Transparent Peer Review file · Nature Communications]

Scalable magnetoreceptive e-skin for energy-efficient high-resolution interaction towards undisturbed extended reality

Corresponding Author: Dr Denys Makarov

Version 0:

Reviewer comments:

Reviewer #1

(Remarks to the Author)

The manuscript by Makushko et al. reports a large-area continuous magnetic sensing fields with low energy consumption, transparency, mechanical compliance, and vapor/liquid permeability. The authors introduce that magnetoreceptor can offer continuous spatial detection of magnetic stimuli across areas up to 120×120 mm with a resolution below 1 mm and is capable for potential applications in virtual reality and augmented reality. I believe that the contribution and significance of this article, as well as the technological progress achieved, have not reached the level that can be published in Nature Communications and would be as appropriately published in another journal for the following reasons.

1. The magnetoreceptor sensor is introduced as E-skin, but the sensor lacks data in the Z-axis direction, thus it is unable to detect pressure. As a result, the sensor can only be called as a trajectory recorder rather than an E-skin.
2. In the practical application of operating the device, a strong magnet have to be worn on the finger, which is not convenient for the daily use and causes discomfort to the human body due to its impermeability. As a result, the application scenario is limited.
3. As shown in Figure 2 b,c and Figure 4, the signal acquisition is based on single channel. The manuscript fails to show whether accurate reading and localization for the short periods of multi-signal stimuli can be promised when signals are applied at two or more different positions simultaneously.
4. The manuscript mentions a resolution of 1mm, but the article does not explicitly indicate an accurate resolution data. The authors may argue that Supplementary Figure 6 is evidence showing its high resolution. In fact, the minimum radius of the circle is 7mm rather than 1mm, so the claim of 1mm-resolution is not supported.
5. There is the movement of the electrodes' position during the stretching process as shown in Supplementary Figure 12. When the device adheres to the skin, the position of the electrodes will disengage its original position, which may cause incorrect identification during the operation. The manuscript fails to show an effective method to avoid such an issue and achieve precision imaging during the experiment.
6. In Figure 5, the authors exhibit a touchless application of the contact lens. Although the authors mentioned "the transparency of the mesh does not impair the original optical functionality". However, in the video showing the application of the Magnetoreceptive contact lens, the lens is placed on the table rather than in human eyes. Since the magnetoreceptor can only be activated by magnets within 1cm, the field of view will get affected due to the electromagnetic pen during operation over the device grid.
7. As a trajectory detection technology rather than pressure distribution sensing, similar work has been extensively studied several years ago, and the results are even more exciting, such as "Self-Powered High-Resolution and Pressure-Sensitive Triboelectric Sensor Matrix for Real-Time Tactile Mapping". As a result, trajectory recognition and low energy consumption can not be recognized as strong innovation points.

Reviewer #2

(Remarks to the Author)

The paper introduces a large-area magneto-receptive skin combining the giant magnetoresistance effect and electrical resistance tomography. The skin provides a high resolution of less than 1 mm and a low power consumption. At the same time, the flexible skin is optically semi-transparent and vapor permeable. Overall, this paper offers a combination of several existing concepts, together with a unique continuous GMR layer, in an unprecedented way, leading to a very attractive e-skin solution.

The English and scholarly presentations are very good, and the paper is well structured with clear explanations of the methods and results.

I propose to accept this manuscript with minor changes:

- The reported concept of the magnetoreceptive, continuous sensor film provides a high resolution as well as effective function. Compared to typical capacitive proximity sensors, the magnetoresistive system will always need an additional device for the magnetic stimulus. Can the authors elaborate more on practical applications and their advantages?
- How will a magnetic contact lens be connected to the external electronics?
- What is the sensitivity of the skin to mechanical stimuli (strain, pressure,...)?
- Is there any data available on the homogeneity (thicknesses, sensitivity,...) of the GMR stack over the large area?
- Is the system sensitive to the alignment of the magnetic stimulus? In other words, will the alignment of the stimulus cause a change in the detected distance of size?

Reviewer #3

(Remarks to the Author)

The manuscript entitled 'Scalable magnetoreceptive e-skin for energy-efficient high-resolution interaction towards undisturbed extended reality' is written well, presenting an innovative approach to integrating magnetoreceptive capabilities into electronic skins with minimized energy consumption and enhanced spatial resolution. This significant contribution addresses critical limitations regarding the scalability and functionality of electronic skins, simultaneously paving new pathways for human-computer interaction, particularly within the realm of extended reality. I advocate for its publication in Nature Communications after some clarifications and improvements.

1. Authors mentioned that 'Different from conventional point- or matrix-configurations where individual magnetic field sensors are distributed at specific points (Fig. 1a,b), EMRT employs a continuous giant magnetoresistive (GMR) multistack layer sputtered onto expansive flexible foils,' and displayed the EMRT connection scheme in Supplementary Fig. 1. I recommend a very brief description of the working principle of EMRT to enhance comprehension.
2. The authors mentioned the 'Py/Cu GMR effect' in other research on page 8. It would be better to briefly mention the Co/Cu GMR the authors used earlier in the paper.
3. In Fig. 1c, an illustrative diagram of the EMRT-enabled magnetoreceptive e-skin shows the connection between the power and signal reception. However, there seem to be no other connections and external power devices in Fig. 1d. A simple explanation of the form or simplification of external devices would clarify this discrepancy.
4. The methodology for adhering the film to the skin and whether the adhesive affects performance is not discussed. A brief description of the seamless integration methods onto human skin and how the film maintains its functionality upon application should be provided.
5. The film testing, especially the evaluation of vapor permeability, is well organized. Supplementary Figure 12 shows the film flattening back to 67%; apart from GMR, how is the film's performance impacted?

Version 1:

Reviewer comments:

Reviewer #2

(Remarks to the Author)

The authors have sufficiently addressed the comments and, in my opinion, the manuscript should be published.

Reviewer #3

(Remarks to the Author)

The manuscript entitled 'Scalable magnetoreceptive e-skin for energy-efficient high-resolution interaction towards undisturbed extended reality' introduces an innovative method for integrating magneto-inductive functionality into electronic skin. It addresses key limitations in scalability and functionality of electronic skins, paving the way for advanced human-computer interaction, particularly in the realm of extended reality. The authors have responded comprehensively to the relevant questions, supplemented the manuscript with appropriate experiments, and made substantial revisions. Therefore, I endorse the publication of this paper in Nature Communications.

Reviewer #4

(Remarks to the Author)

This manuscript presents a method for creating spatially continuous, high-resolution magnetic field maps using the electrical magnetoresistive tomography (EMRT) technique. This approach is based on an extended magnetoreceptive medium and electrical resistance tomography, eliminating the need for active transistor matrices and intricate sensor setups typically seen in traditional active-matrix systems. The subject of this study is intriguing and holds significant potential. However, the manuscript lacks some fundamental information. Based on the reported results, I can recommend this study for possible publication in Nature Communications if the authors address the following comments:

1. The authors did not provide reliable evidence for the practical application of the magnetoreceptive contact lens. This shortcoming needs to be addressed by demonstrating practical applications or providing detailed experimental evidence.

2. The magnetoreceptor can only be activated by magnets within a 1 cm range, which may impact the field of view when using an electromagnetic pen over the device grid. The authors should explain this limitation and propose possible practical solutions.
3. The wireless technology behind the contact lens system is not clearly explained. The authors should specify whether they incorporate specific wireless technology and, if so, provide details on its implementation and functionality.
4. The authors need to investigate and discuss the biocompatibility of the developed e-skin. It is essential to ensure that all materials used are safe for long-term use on human skin. The manuscript mentions that the magnetoreceptive electronic skin is permeable to vapor, moisture, and sweat, reducing the risk of skin irritation. The authors should provide a detailed analysis and experimental validation of these properties to ensure the material's practical biocompatibility.
5. There is insufficient information on the tactile sensitivity of the e-skin. The authors should provide data on pressure sensitivity, range, and limit of detection (LOD). Additionally, they should include $\Delta R/R_0$ versus applied pressure and demonstrate the tactile response for cyclic applications, with a minimum of 1000 cycles.
6. The reliable performance of the e-skin under different humidity and temperature conditions is unclear. The authors should investigate and report the effects of temperature, pH, and humidity on the e-skin's performance. Have the authors conducted long-term performance tests to assess the durability and stability of the magnetoreceptive electronic skin? It is important to determine if the sensor can accurately detect and localize magnetic input when deformed or stretched over time.

Version 3:

Reviewer comments:

Reviewer #4

(Remarks to the Author)

The manuscript entitled "Scalable magnetoreceptive e-skin for energy-efficient high-resolution interaction towards undisturbed extended reality (NCOMMS-24-15778C)" is interesting, and the quality of the manuscript is sufficient to be considered for publication. Overall, the authors have addressed my comments in the revised manuscript, and I recommend it for possible publication in Nature Communications.

Response letter

We would like to thank the Reviewers for their remarks and suggestions, all of which are addressed in the revised manuscript. All changes in the manuscript are indicated in blue.

Our itemized responses to all the Reviewer's comments are below.

Reviewer #1 (Remarks to the Author)

The manuscript by Makushko et al. reports a large-area continuous magnetic sensing fields with low energy consumption, transparency, mechanical compliance, and vapor/liquid permeability. The authors introduce that magnetoreceptor can offer continuous spatial detection of magnetic stimuli across areas up to 120×120 mm with a resolution below 1 mm and is capable for potential applications in virtual reality and augmented reality. I believe that the contribution and significance of this article, as well as the technological progress achieved, have not reached the level that can be published in Nature Communications and would be as appropriately published in another journal for the following reasons.

1. "The magnetoreceptor sensor is introduced as E-skin, but the sensor lacks data in the Z-axis direction, thus it is unable to detect pressure. As a result, the sensor can only be called as a trajectory recorder rather than an E-skin."

Answer: We respectfully disagree with the Reviewer that our device does not fit the term e-skin, as it lacks sensitivity to mechanical pressure. In fact, the state-of-the-art flexible e-skins incorporate numerous sensing elements to detect a variety of stimuli beyond just pressure, including strain (DOI: 10.1002/adfm.202007495), shear force (DOI: 10.1038/s41467-020-19531-0; 10.1002/adfm.202104686), temperature (DOI: 10.1002/adma.201504659), humidity (DOI: 10.1016/j.nanoen.2022.107077), light (DOI: 10.1038/s41467-017-02685-9), chemical and biochemical (DOI: 10.1002/adma.201302240; 10.1039/C8TB02862A), and magnetic fields (DOI: 10.1002/aelm.202300677). The magnetic field proximity sensing reported in our paper is designed to expand the functionality of e-skins towards perceiving magnetic fields enabling touchless interactivity rather than to replace the pressure (touch) sensing.

The magnetoreceptor proposed in the paper relies on the giant magnetoresistive effect, which is a classical magnetic proximity sensor [DOI: 10.1016/j.sna.2023.114500; 10.1021/acsnm.3c01936]. In the scope of this research, the sensitivity in the Z-axis direction is defined as proximity sensing. The Z-axis sensitivity of our magnetoreceptor is a measure of vertical distance between the magnetic source (e.g., magnetic stylus, magnetic skin) and the sensor plane, as discussed in Figure 2f,g of the main text and in Supplementary Figure 4c,e. The potential applications of Z-axis sensitivity of our magnetoreceptor are discussed in Figure 5. Here, different magnitudes of the electrical magnetoresistive tomography (EMRT) signals (i.e., a Z-axis distance dependent parameter) are used to determine the function executed by the software: Navigation (medium signal) and Selection (high signal).

2. *“In the practical application of operating the device, a strong magnet have to be worn on the finger, which is not convenient for the daily use and causes discomfort to the human body due to its impermeability. As a result, the application scenario is limited.”*

Answer: The functional elements for magnetic field sensing used in this research are based on the giant magnetoresistance (GMR) effect of multi-stack Cu/Co layers. The GMR sensors have high magnetoresistance change (Figure 1i, Supplementary Figure 3), that facilitates a precise detection of small magnetic fields (DOI: [10.1103/PhysRevLett.61.2472](https://doi.org/10.1103/PhysRevLett.61.2472); [10.1103/PhysRevB.39.4828](https://doi.org/10.1103/PhysRevB.39.4828)). Thanks to the high sensitivity of GMR sensors, our magnetoreceptors can detect weak magnetic fields, e.g., generated from soft magnetic composites (a mixture of polymeric elastomer and permanent magnet fillers) used as magnetic skins. To test the sensing capability of our magnetoreceptor, we conducted additional experiments where soft composites, consisting of NdFeB magnetic powders and polydimethylsiloxane (PDMS), interacted with the large-area transparent magnetoreceptor. As shown in Supplementary Movie 6 and Supplementary Figure 7, our magnetoreceptor successfully recognizes input via magnetic skins. The high sensitivity facilitates the extensive incorporation of the large-area transparent magnetoreceptor into various application scenarios by eliminating the need for bulky strong magnets. Indeed, such magnetic composites, featuring mechanical properties similar to human skin, have been widely reported and used as magnetic skins for human-machine interaction (DOI: [10.1002/admt.201900493](https://doi.org/10.1002/admt.201900493); [10.1038/s41563-021-01093-1](https://doi.org/10.1038/s41563-021-01093-1); [10.1002/aisy.201900025](https://doi.org/10.1002/aisy.201900025)).

“When utilizing a magnetic stylus (mechanical pencil filled with NdFeB permanent magnets) or similar input devices (e.g., magnetic skins consisting of polymeric elastomers and permanent magnet powders) for magnetic stimuli, the magnetic stray fields lead to localized drops in the resistance of the magnetoreceptor (Fig. 1j,k and Supplementary Figs. 5-8).”, as stated on page 6 of the revised manuscript. Also, a new Supplementary Movie 6 and Supplementary Figure 7 (page 11 in revised Supplementary Materials) are introduced to address the comment of the referee.

Supplementary Figure 7. Two-point interaction with the EMRT conditioned large-area magnetoreceptor. (a) A selected frame from the Supplementary Movie 6 showing two mechanically flexible magnetic skins (labelled as “I” and “II”) attached to fingertips of a user. (b) Images of the magnetic stray field profiles measured at the magnetic skin surface. The measurement of stray fields is carried out using cmos-magview device (matesy GmbH, Germany). (c,f) Selected frames of the Supplementary Movie 6 showing a single-point and two-point interactions with the magnetoreceptive mesh using magnetic skin on fingertips. (d,g) The corresponding EMRT reconstruction of the magnetoresistive response and (e,h) double Gaussian fit analysis of the data allowing to locate the position of each magnet. In panel (e) only one Gaussian peak is shown as the second one is of low intensity (fingertip with the magnetic skin II is lifted away from the sensor surface).

3. “As shown in Figure 2 b,c and Figure 4, the signal acquisition is based on single channel. The manuscript fails to show whether accurate reading and localization for the short periods of multi-signal stimuli can be promised when signals are applied at two or more different positions simultaneously.”

Answer: To address the comment of the Reviewer, we performed additional experiments to show the capability of our magnetoreceptors in accurate reading and localization of multi-signal stimuli. To enable multi-channel acquisition, we utilized similar data-processing

principles, albeit with more advanced algorithms, to identify and track various input stimuli. Supplementary Figures 7,8 and Movies 6,7 validate that the magnetoreceptors can interact with multiple (2 or even 4) magnetic inputs concurrently.

In the first experiment, the magnetoreceptor engaged with two soft magnetic skins—composed of PDMS and NdFeB powder affixed to the fingertips of a user. The EMRT algorithm effectively elucidated single- and two-point interactions (see Supplementary Figure 7 and Movie 6). To locate both magnetic inputs simultaneously, a numerical calculation involving double-Gaussian fitting was executed. This method reliably operated even when the two magnetic inputs were positioned closely, as a single broad spot in the EMRT image can be readily decomposed into two distinct input signals. Please see the figure attached to the response to the Reviewer's comment 2.

To extend the multi-channel acquisition capability, we augmented the stimuli quantity. Supplementary Figure 8 and Movie 7 demonstrate that our magnetoreceptor is able to discern three or even four stimuli concurrently (represented with four cubic magnets), affirming its ability to deliver accurate reading and localization for the short periods of multi-signal stimuli when signals are applied at two or more different positions simultaneously.

Supplementary Figure 8. Identifying multiple magnetic inputs. (a,b) Selected frames of the Supplementary Movie 7 showing the experimental setup. A large-area magnetoreceptor is placed above the steel plate and is used to locate multiple cube-shaped permanent magnets. The corresponding EMRT reconstructed magneto-resistive response is shown on a computer screen and in panels (c,d). The positions of four magnets are reconstructed as peaks on the resistance maps. The magnetoreceptor was capped with 125- μm -thick PET foil to prevent shortcutting onto the metallic surface of the magnets.

The following new information is added to the revised manuscript on page 16. “Notably, our magnetoreceptor allows for accurate reading and localization of multi-channel magneto-electrical signals when magnetic stimuli are applied at two or more different positions simultaneously (please refer to Supplementary Figs. 7,8 and Supplementary Movies 6,7). The multi-channel reading capability facilitates the development of more advanced interactive devices.”

4. “The manuscript mentions a resolution of 1mm, but the article does not explicitly indicate an accurate resolution data. The authors may argue that Supplementary Figure 6 is evidence showing its high resolution. In fact, the minimum radius of the circle is 7mm rather than 1mm, so the claim of 1mm-resolution is not supported.”

Answer: To confirm the fact of 1 mm resolution, we evaluate our magnetoreceptor from two aspects:

1) We place the magnetic stylus at a specific point and then check the deviation of the EMRT-reconstructed position from its actual coordinate.

The spatial acuity of the magnet positioning in the XY-plane (denoted as spatial resolution) is qualitatively analysed in Figure 2g of the main text. In this experiment, the magnetic stylus approached the magnetoreceptor plane at fixed position (x,y) (Supplementary Figure 12). And the deviation of the EMRT reconstructed position from its real coordinate ($x_0=x-x_i$; $y_0=y-y_i$) is used to estimate the evolution of spatial resolution with distance of the magnetic stylus relative to the magnetoreceptor plane (Z-axis coordinate). As the stylus is within 10 mm from the sensor plane, the deviation of the EMRT reconstructed position is within about 1 mm (Supplementary Figure 12). From these results, we conclude that the spatial resolution is 1 mm.

The corresponding information from the Figure 2g of the main text is duplicated and extended as new Supplementary Figure 12 (on page 16 of revised Supplementary Materials).

Supplementary Figure 12. Accuracy of the spatial acuity of the EMRT conditioned magnetoreceptor. (a) Magnetic stylus is positioned above a large-area magnetoreceptor at a fixed (x,y) position and is vertically moved away (b) from the sensor plane (along z-direction). (c) The deviation of the EMRT reconstructed stylus position from the (x,y) with the increasing distance along z-axis. (d) A zoomed in region of the panel (c) showing the

deviation of the reconstructed position from the actual position of being less than 1 mm as the distance from the sensor plane is less than 10 mm.

2) We manually drew circles of different diameters and then compared the average deviation of the EMRT reconstructed patterns from the ideal trajectory. This average *deviation*, rather than the circles dimensions, can be used to evaluate the sensing resolution.

For all circles, the average deviations of EMRT reconstructed patterns from the ideal trajectory were no more than 1 mm (Supplementary Figure 10). The increased deviations for larger circles may stem from the inaccuracies of human hand drawing. The results confirm the spatial resolution of our magnetoreceptor at the level of about 1 mm.

The corresponding information is updated in the legend to Supplementary Figure 10.

Supplementary Figure 10. A set of concentric circles hand-drawn using the EMRT based touchless interaction setup. The mesh size is 70 x 70 mm², EMRT is exploited in 16-contacts probe geometry and 1 mA probing current. Lines represent the intend-to-draw circles and symbols stand for experimental result. A circle of a 7 mm diameter is reliably drawn. The average displacement between the recorded and intended trajectory is within a limit of 1 mm. A larger average displacement of 1.4 mm at the largest drawn circle is related to the human factor, as the shapes were drawn by hand. The result indicates high spatial resolution of the EMRT method below 1 mm.

5. *“There is the movement of the electrodes’ position during the stretching process as shown in Supplementary Figure 12. When the device adheres to the skin, the position of the electrodes will disengage its original position, which may cause incorrect identification during the operation. The manuscript fails to show an effective method to avoid such an issue and achieve precision imaging during the experiment.”*

Answer: The reviewer is right with this observation. In fact, this is a common problem faced by all areas of stretchable electronics. In our case, the stability of the contacts was absolutely sufficient to conduct the reliable measurement. This is achieved by realising rather extended contact pads, which can slide over the conducting surface of the sensor. As the sensor is continuous but not an array of individual sensing elements, this sliding does not affect the readout as can be clearly seen from the reported resistance measurements. This approach to realize sufficiently stable contacts to stretchable electronics is one of many ways to solve the contacting problem for stretchable electronics. To this end, there are many solutions reported, for example, forming wavy configuration on compliant supports (as detailed in DOI: 10.1002/adma.200600646, 10.1073/pnas.0702927104, 10.1038/nnano.2006.131, 10.1126/science.1121401), designing serpentine interconnects (as reported in DOI: 10.1039/C3SM51360B, 10.1002/adfm.201302957, 10.1002/adma.201604989), or generating two-dimensional/three-dimensional spiral interconnects (as developed in DOI: 10.1063/1.4898128, 10.1016/j.eml.2014.12.008), and so on. For more information on realizing stretchable electronics and operating during stretching, see numerous review articles (DOI: 10.1126/science.1182383, 10.1002/adma.201902254, 10.1002/adfm.201504755, 10.1002/adma.201302240, 10.1021/acsnano.7b04898, 10.1002/adma.201504366, 10.1002/adma.201904765, 10.1039/C7CS00730B, 10.1002/adma.201303349, 10.1002/sml.201602790). We note that some of these approaches can be also applied to condition our large-area magnetoreceptors proven that the current contact design will be not reliable.

6. *“In Figure 5, the authors exhibit a touchless application of the contact lens. Although the authors mentioned “the transparency of the mesh does not impair the original optical functionality”. However, in the video showing the application of the Magnetoreceptive contact lens, the lens is placed on the table rather than in human eyes. Since the magnetoreceptor can only be activated by magnets within 1cm, the field of view will get affected due to the electromagnetic pen during operation over the device grid. ”*

Answer: Thank the referee for the comments. We will address them point-by-point.

“In Figure 5, the authors exhibit a touchless application of the contact lens. Although the authors mentioned “the transparency of the mesh does not impair the original optical functionality”. However, in the video showing the application of the Magnetoreceptive contact lens, the lens is placed on the table rather than in human eyes.”

We agree on the comment *“the lens is placed on the table rather than in human eyes”*. Although it would be appealing to conduct experiments on human eyes, we do not have permission for carrying out this experiment on humans. Our work highlights the development of transparent and imperceptible magnetoreceptors, achieving continuous sensing of magnetic

fields across large areas with high sensing resolution and low energy consumption. The smart contact lens is just one of numerous potential applications for these unique magnetoreceptors. We will keep this suggestion in mind and seek collaboration from researchers with licenses for such biological experiments, or try to obtain the necessary permissions ourselves to confirm the utility of this application with human eyes. However, either approach will take a very long time. We sincerely ask the Reviewer for understanding that this task is not performed in this study.

“Since the magnetoreceptor can only be activated by magnets within 1cm, the field of view will get affected due to the electromagnetic pen during operation over the device grid.”

This issue can be addressed in two ways. First, the magnetic pen can be replaced with magnetic skins made of mechanically soft magnetic composites, as confirmed in our responses to Question 2 of the Reviewer. These magnetic skins can be easily attached to fingers, serving as a source of magnetic stimuli and reducing the impact on the field of view caused by the magnetic pen during operation. Using magnetic skins on a finger has another advantage: Since two eyes provide stereo vision, partial obstruction of the view of one of the eyes should not cause significant discomfort during interaction with the device.

The mentioned working distance of 10 mm, reported in the manuscript, is a conservative estimation that provides high spatial resolution for the magnet positioning (please, see Fig. 2f,g of the main text). The value can be further enhanced by changing the magnetoreceptive medium [e.g., instead of Co/Cu GMR multilayers it is possible to use more sensitive Py/Cu GMR sensors (please see the newly added Supplementary Figure 13) or even sensors relying on the tunnelling magnetoresistance effects (DOI: [10.1063/1.3300717](https://doi.org/10.1063/1.3300717); [10.1002/adma.201602910](https://doi.org/10.1002/adma.201602910); [10.1038/ncomms7080](https://doi.org/10.1038/ncomms7080))] as well as use of more sophisticated data processing algorithms capturing the gradient of the resistance change over the magnetoreceptor area.

Supplementary Figure 13. Magnetoresistive performance of [Cu/Cu]₅₀ and [Py/Cu]₅₀ multilayer stacks. Typical magnetoresistive curves (a,c) and their first derivatives (b,d) that represent performance characteristics of magnetoreceptive elements. The Co/Cu multilayers reveal higher GMR ratio of 25 % compared to Py/Cu multilayer (with a typical GMR magnitude of about 7 %). However, the Py/Cu sensors reveal higher sensitivity, reaching 2%/T and are characterized by a smaller saturation field. These differences arise from the stronger magnetic anisotropy and stronger electron scattering of cobalt, as well as the soft magnetic properties and weak electron scattering behavior of permalloy. From the application point of view, this information gives a clue that the composition of the GMR layers (Py-based or Co-based) should be optimized for the specific needs. Permalloy-based magnetoresistive elements are more suitable for measuring weak magnetic signals, while cobalt-based sensors may be used in more general scenarios requiring a larger magnetic field range.

7. “As a trajectory detection technology rather than pressure distribution sensing, similar work has been extensively studied several years ago, and the results are even more exciting, such as “Self-Powered High-Resolution and Pressure-Sensitive Triboelectric Sensor Matrix for Real-Time Tactile Mapping”. As a result, trajectory recognition and low energy consumption can not be recognized as strong innovation points.”

Answer: We kindly disagree with this comment. In particular, we emphasise that our work is not just “a trajectory detection technology”. In this research, we fabricated wafer-scale magnetoreceptors with optical transparency, mechanical compliance, and vapor/liquid permeability, which allows for its imperceptible integration on skin. These magnetoreceptors achieve continuous sensing of magnetic fields across an area of $120 \times 120 \text{ mm}^2$ with a sensing resolution of better than 1 mm by incorporating the giant magnetoresistance effect and electrical resistance tomography. More importantly, our technical innovation breaks through the typically quadratic dependence of energy consumption and circuit complexity on the sensing area, enabling magnetoreceptors with three orders of magnitude less energy consumption when compared to state-of-the-art transistor-based magnetosensitive matrices. Because of these achievements, our magnetoreceptors open the door to a variety of exceptional applications, e.g., on-skin XR interfaces and contact lenses as exemplarily demonstrated in our manuscript. The trajectory detection is just one functionality realized with our magnetoreceptors in numerous applications.

About the comparison of our technique (i.e., *touchless* magnetic sensing) with “*pressure distribution sensing*”, it is not a matter of which technology is better or worse. There is no one-size-fits-all answer and it really depends on the specifics of the applications and situations. Take the paper as an example: “*Self-Powered High-Resolution and Pressure-Sensitive Triboelectric Sensor Matrix for Real-Time Tactile Mapping*” cited by the Reviewer. It reports large-area pressure sensing with triboelectric sensor matrices, which relies on direct contact between objects and the triboelectric sensor. In contrast, our devices utilize the magnetoresistance effect for *touchless* sensing. Through magnetic interactions, magnetic objects can be sensed remotely without the need for direct physical contact, offering promising potentials for touchless applications where physical contact is impossible or undesirable. For instance, in the medical field or public places, physical contact might cause the spread of bacteria and viruses, posing health risks to doctors, patients and other users. Additionally, our magnetoreceptors exhibit several other properties: optical transparency, mechanical conformability, and vapor/liquid permeability, which further expand the application areas. In fact, the two works are exploring completely different techniques and it would be unfair to compare them directly. We believe these are two complementary technologies that, when combined, can significantly improve the performance of e-skins.

Reviewer #2 (Remarks to the Author)

“The paper introduces a large-area magneto-receptive skin combining the giant magnetoresistance effect and electrical resistance tomography. The skin provides a high resolution of less than 1 mm and a low power consumption. At the same time, the flexible skin is optically semi-transparent and vapor permeable. Overall, this paper offers a combination of several existing concepts, together with a unique continuous GMR layer, in an unprecedented way, leading to a very attractive e-skin solution. The English and scholarly presentations are very good, and the paper is well structured with clear explanations of the methods and results. I propose to accept this manuscript with minor changes.”

Answer: We thank the Reviewer for his/her positive assessment of our manuscript. In the following, we will itemize our responses to the comments.

1. The reported concept of the magnetoreceptive, continuous sensor film provides a high resolution as well as effective function. Compared to typical capacitive proximity sensors, the magnetoresistive system will always need an additional device for the magnetic stimulus. Can the authors elaborate more on practical applications and their advantages?”

Answer: It is indeed true that *“Compared to typical capacitive proximity sensors, the magnetoresistive system will always need an additional device for the magnetic stimulus.”* Luckily, because of the giant magnetoresistance and high sensitivity inherent in giant magnetoresistive sensors, our magnetoreceptors are capable of detecting weak magnetic fields and small magnetic field variations. As a result, the magnetic stimulus can be generate by magnetic composites (i.e., made of soft polymers and permanent magnetic powders) that are compliant to human skins and have been widely used as magnetic skins (DOI: [10.1002/admt.201900493](https://doi.org/10.1002/admt.201900493); [10.1038/s41563-021-01093-1](https://doi.org/10.1038/s41563-021-01093-1); [10.1002/aisy.201900025](https://doi.org/10.1002/aisy.201900025)). Supplementary Movie 6 and Supplementary Figure 7 confirms that the magnetic skins are able to produce magnetic fields that are strong enough to sustain information exchange with the magnetoreceptor. The incorporation of soft magnetic skins could eliminate the dependence on magnetic stylus, facilitating the wide application of the magnetoreceptors.

Compared to the electrical coupling effects used in capacitive proximity sensors, which are susceptible to surrounding interferences such as water droplets and obstacles, the magnetic interactions employed by our magnetoreceptor are immune to these factors. This robust functionality ensures that our sensors operate effectively in a wide range of conditions. For example, our magnetoreceptor can function in aqueous environments, e.g., on rainy days or in swimming pools, which is very challenging for capacitive proximity sensors. Additionally, traditional e-skins made with capacitive proximity sensors experience reduced performance or malfunction when covered by barriers. In contrast, our magnetoreceptor can function not only on bare skin but also through garment, gloves, or other obstacles, providing users with greater freedom for uninterrupted and versatile interactions.

In addition, in situations where electrical coupling is not easily excited, capacitive proximity sensors may fail to function properly. For example, a disabled person, wearing prosthetic

fingers normally made of insulating polymers, will find it very difficult to operate capacitive proximity sensors. In contrast, magnetic skins can be easily incorporated into prosthetic fingers, enabling effortless manipulation of the magnetoreceptors.

It's worth noting that it is not a matter of one technique being better than the other. Both techniques have their own advantages/disadvantages. If effectively combined, the two technologies can complement each other, leading to more versatile e-skins.

The above discussions are summarized in the revised manuscript, as follows:

“When utilizing a magnetic stylus (mechanical pencil filled with NdFeB permanent magnets) or similar input devices (e.g., magnetic skins consisting of polymeric elastomers and permanent magnet powders) for magnetic stimuli, the magnetic stray fields lead to localized drops in the resistance of the magnetoreceptor (Fig. 1j,k and Supplementary Figs. 5-8).”, on page 6 of the revised manuscript.

“Compared with traditional interactive methods (e.g., through optical, touch, capacitive sensing), our magnetoreceptors offer unique advantages in XR, i.e., remaining functionally unaffected due to magnetic immunity to disruptions caused by sweat, moisture, or opaque objects⁴⁶⁻⁴⁸. Such extensive adaptability allows operation not only on bare skin for sensing and tracking, but also through garment or other barriers, even in swimming pools or on rainy days, that typically hinder traditional on-skin sensing systems, providing users with uninterrupted and versatile ways of interaction with digital worlds.”, on page 16 of the revised manuscript.

“Further, the robust touchless interaction enabled by our magnetoreceptor can effectively extend the perceptual range for people with disabilities. For instance, blind individuals equipped with the magnetoreceptor system could extend the distance of their perception, and those with prosthetic hands could use it to interact with smartphones, overcoming the challenge of insulated prosthetics not being able to interact with capacitive touchscreens.”, on page 20 of the revised manuscript.

2. How will a magnetic contact lens be connected to the external electronics?

Answer: The state of the art smart contact lenses include wireless power management and data transfer (DOI: 10.1126/sciadv.aba3252, DOI: 10.1126/sciadv.aap9841, DOI: 10.1002/admt.201900728). Concepts of on-site energy harvesting were reported as well (DOI: 10.1002/sml.202401068). Thus, the magnetoreceptive lens can potentially be run using wireless interface. Such devices will especially benefit from the low power consumption of the EMRT driven magnetoreceptor, which can be even further scaled down compared to values reported in the paper (Supplementary Table S2) by implementing only 8-electrodes readout geometry (Supplementary Movie 3). In this case, the amount of measure iterations per EMRT cycle is reduced to 40, also lowering the amount of information that has to be transferred to the external device for processing and tomography reconstruction. For example, in combination with magnetic skins (DOI: 10.1002/adem.202000944) this can be used to realize eye tracking in sleep etc.

3. *What is the sensitivity of the skin to mechanical stimuli (strain, pressure, ...)?*

Answer: We will address the comment by investigating the sensitivity of the skin to three types of mechanical stimuli (including bending, pressure, and strain).

Figure 3c and Supplementary Fig. 16 indicate that the magnetoreceptor has reliable flexibility. Even when bent to a radius of curvature of about 250 μm , the magnetoreceptor maintains stable electrical resistance and magnetoresistance. The mechanical stability may arise from the sandwich configuration, where the active magnetoresistive film is enclosed between two layers of flexible polymers and the polymer foils serve as a scaffold to protect the thin magnetoresistive films from mechanical damage.

Unlike printed devices where electrical conductive pathways are established by physical contacts between functional fillers, making them susceptible to mechanical pressure, our magnetoreceptors are composed of Cu/Co multi-stack layers. Their compact and continuous configuration is beneficial for maintaining the electrical performance under mechanical pressure. Supplementary Video 7 confirms this expectation, showing that pressuring with a finger does not cause any signal noise into the working magnetoreceptor.

To endow the magnetoreceptor, constructed from a Cu/Co metallic film, with the ability to stretch, we attach it to a pre-stretched viscoelastic tape. Upon releasing the pre-stretch, the magnetoreceptor acquires the stretching property. Our findings indicate that our magnetoreceptor can endure up to 100% uniaxial and 75% biaxial elongation without significant changes in magnetoresistance (Figure 3d, Supplementary Figs. 17,18).

4. *Is there any data available on the homogeneity (thicknesses, sensitivity, ...) of the GMR stack over the large area?*

Answer: Following the Reviewer's suggestion, we sputtered the GMR stack onto a 300 mm diameter wafer and tested the electrical resistance and magnetoresistance performance at three areas: one is the central point and the other two are 90 mm and 140 mm away from the center. We confirm experimentally that the electrical resistance varies less than 5 %. Furthermore, the magnetoresistance was almost identical across the three areas. The stable magnetoresistance performance of the sputtered GMR stack is desirable for maintaining a uniform magnetic response of the magnetoreceptor over large areas. The calibration of the homogeneity of the GMR stack over the large wafer area is summarized in Supplementary Figure 3 on page 7 of the revised Supplementary Materials:

Supplementary Figure 3. Spatial homogeneity of the Co/Cu GMR multilayers. (a) Magnetoresistive response of the GMR layer deposited onto a 300-mm-diameter silicon wafer. The legend corresponds to the radial position from the center of the wafer, as depicted in panel (b). The GMR performance is homogeneous throughout the whole surface, while a slight increase in the resistance (about 5%) is observed at the very edge. Sample resistance measured at zero magnetic field in the center of the wafer (position indicated as “0”) is 5.19 Ohm. Sample resistance measured at the location, which is 90 mm away from the center of the wafer (position indicated as “90”), is 5.15 Ohm. Sample resistance measured at the edge of the sample, which is 140 mm away from the center of the wafer (position indicated as “140”), is 5.40 Ohm.

5. *Is the system sensitive to the alignment of the magnetic stimulus? In other words, will the alignment of the stimulus cause a change in the detected distance of size?*

The sensitivity of our device to the alignment of the magnetic stimulus is determined by the two factors: (i) the Co/Cu GMR multilayers are predominantly in-plane sensitive, so the higher sensitivity is expected when the magnetic stimulus is oriented in such a way that the stay fields will be within the sensor plane. (ii) The magnetic fields generated by the magnetic stylus or similar sized magnets are strongly inhomogeneous and rapidly decay with distance (Supplementary Figure 5). Thus, higher response can be expected when the magnetic device is pointing with magnetic pole towards the sensor plane.

To address the comment of the Reviewer, we have performed an experiment investigating the preferred orientation of the magnetic input relative to the large-area magnetoreceptor plane. In the experiment, the magnetoreceptor (70x70mm² in size) is covered by a 5-mm-thick PMMA sheet and a cube-shaped permanent magnet is placed on top. The poles of the magnet are marked with red and blue colours. The magnetoreceptor is sequentially approached by a differently oriented magnet (panels (b-e) of Supplementary Figure 6). It can be seen, that higher EMRT response, and thus higher resistance change, is observed when the magnet is facing pole towards the magnetoreceptor plane. When the magnet is positioned sideways (poles are horizontal), the electrical response is halved. Therefore, the magnetic input device

should be magnetized with pole facing the magnetoreceptor plane to benefit from a stronger sensor signal. The corresponding information is introduced as Supplementary Figure 6 at the page 10 of the revised Supplementary Materials.

Supplementary Figure 6. Sensitivity of the EMRT conditioned magnetoreceptor to the magnet orientation. (a) A schematics of the experiment: a cube-shaped permanent magnet with marked poles (North – red and South – blue) is placed above the large-area magnetoreceptor. A 5-mm-thick PMMA sheet is used as a spacer between the magnet and the magnetoreceptor. (b-e) Photographs of the bar magnet placed above the magnetoreceptor with different orientation of its poles and (f-i) the corresponding reconstructed resistance maps. All panels are plotted using the same color scaling. (j) The EMRT signal timeline recorded during the experiment. The EMRT reconstructed GMR response of the magnetoreceptor is twice smaller when the magnet is placed on a side, as compared to when the magnet is facing the pole towards the magnetoreceptor plane. This is related to the higher magnetic flux close to the poles of the magnet compared to its sides.

Reviewer #3 (Remarks to the Author)

“The manuscript entitled ‘Scalable magnetoreceptive e-skin for energy-efficient high-resolution interaction towards undisturbed extended reality’ is written well, presenting an innovative approach to integrating magnetoreceptive capabilities into electronic skins with minimized energy consumption and enhanced spatial resolution. This significant contribution addresses critical limitations regarding the scalability and functionality of electronic skins, simultaneously paving new pathways for human-computer interaction, particularly within the realm of extended reality. I advocate for its publication in Nature Communications after some clarifications and improvements.”

Answer: We appreciate the positive remarks of the Reviewer. In the following, all comments will be addressed point-by-point.

1. *“Authors mentioned that ‘Different from conventional point- or matrix-configurations where individual magnetic field sensors are distributed at specific points (Fig. 1a,b), EMRT employs a continuous giant magnetoresistive (GMR) multistack layer sputtered onto expansive flexible foils,’ and displayed the EMRT connection scheme in Supplementary Fig. 1. I recommend a very brief description of the working principle of EMRT to enhance comprehension.”*

Answer: Following the suggestion of the Reviewer, we have introduced a section in Methods that provides a short description of the realized EMRT concept:

Electrical Magnetoresistive Tomography (EMRT): A square-shaped large-area transparent GMR magnetoreceptor is electrically connected along its periphery with 16 evenly distributed electrodes (4 electrodes per side). The number of electrodes and their positioning can vary according to specific requirements and magnetoreceptor geometry. The electrical sampling is done following the algorithm: an electrical current is sequentially injected through the pairs of adjacent electrodes and the resulting voltages are measured between other pairs of neighboring electrodes (Supplementary Fig. 4). The sequence is repeated until all the electrode pairs are probed and the corresponding voltage dataset is collected (Supplementary Table S1). This process is driven by a custom-built switching matrix module composed of PXI 2535 matrix switch and PXI3-4303 analog input modules (National Instruments, USA) (Supplementary Fig. 1) controlled by a LabVIEW (18.0f2 (64-bit)) program. Next, the measured voltage dataset is used to reconstruct the electrical resistance (conductivity) map. This is done using the EIDORS (Electrical Impedance Tomography and Diffuse Optical Tomography Reconstruction Software)⁴⁹, an open and free software algorithms for forward and inverse modeling for Electrical Impedance Tomography driven by Matlab (R2018b (64-bit)). The reconstruction is performed in the differential mode, as this approach allows to isolate the GMR-induced resistance change from unwanted external influences (such as uneven resistance of contact points). For this, the first measured dataset is used to account for the baseline of the resistivity map and the subsequent measurements show the local GMR response induced by a magnetic trigger, e.g. magnetic stylus or magnetic skin. During the reconstruction, a 2D finite element model (FEM) of the magnetoreceptor is built (Fig. 1g) and

the resistivity map is obtained using an iterative Gauss-Newton algorithm⁴⁹. The obtained resistivity map is then post-treated according to the pre-defined use-case scenario and visualized using Wolfram Mathematica script. If only one magnetic input is anticipated, the position of the magnetic stimulus is attributed to the FEM element with the highest conductivity change. A more advanced analysis is required to recognize multiple simultaneous magnetic inputs. In this work, it is realized using multi-Gauss fit of the reconstructed resistivity map. Spatial positions of each peak is interpreted as interaction point (Supplementary Fig. 7 and Supplementary Movie 6).

2. *“The authors mentioned the ‘Py/Cu GMR effect’ in other research on page 8. It would be better to briefly mention the Co/Cu GMR the authors used earlier in the paper.”*

Answer: We thank the Reviewer for this suggestion. Permalloy/Cu GMR sensors differ significantly from Co/Cu GMR sensors in terms of magnetoresistance magnitude, saturation magnetic field, and sensitivity. Co/Cu sensors, due to a stronger magnetic anisotropy and stronger electron scattering of cobalt, exhibit higher saturation magnetic fields and greater changes in magnetoresistance but somewhat lower sensitivity at low magnetic fields (Supplementary Figure 13). In contrast, permalloy/Cu sensors, due to the soft magnetic nature and weaker electrons scattering of permalloy, have lower saturation magnetic fields and higher sensitivity, especially at low magnetic fields, although their changes in magnetoresistance are smaller (Supplementary Figure 13).

To address the comment, we performed additional experiments to characterize the magnetoresistive performance of Permalloy/Cu GMR sensors. The comparison between Py/Cu GMR sensors and Co/Cu GMR sensors have been added on page 17 of the revised Supplementary Information (Supplementary Figure 13).

Supplementary Figure 13. Magnetoresistive performance of [Cu/Cu]₅₀ and [Py/Cu]₅₀ multilayer stacks. Typical magnetoresistive curves (a,c) and their first derivatives (b,d) that represent performance characteristics of magnetoreceptive elements. The Co/Cu multilayers reveal higher GMR ratio of 25 % compared to Py/Cu multilayer (with a typical GMR magnitude of about 7 %). However, the Py/Cu sensors reveal higher sensitivity, reaching 2%/T and are characterized by a smaller saturation field. These differences arise from the stronger magnetic anisotropy and stronger electron scattering of cobalt, as well as the soft magnetic properties and weak electron scattering behavior of permalloy. From the application point of view, this information gives a clue that the composition of the GMR layers (Py-based or Co-based) should be optimized for the specific needs. Permalloy-based magnetoresistive elements are more suitable for measuring weak magnetic signals, while cobalt-based sensors may be used in more general scenarios requiring a larger magnetic field range.

3. “In Fig. 1c, an illustrative diagram of the EMRT-enabled magnetoreceptive e-skin shows the connection between the power and signal reception. However, there seem to be no other connections and external power devices in Fig. 1d. A simple explanation of the form or simplification of external devices would clarify this discrepancy.”

Answer: Figure 1d depicts the concept of magnetoreceptive e-skin adhered onto a wrist of a person and its application as a human-machine interface for interaction with a smartphone. We experimentally realized this concept, as evidenced in Supplementary Figure 9 and Supplementary Movie 1. In both, the contact points distributed along the perimeter of the magnetoreceptor can be observed. To address the comment of the Reviewer, we updated the legend of Figure 1 of the main text to avoid misunderstanding and modified the Supplementary Figure 9, highlighting the electrical contacts positions.

Supplementary Figure 9. Magnetoreceptive permeable GMR mesh for on-skin applications. (a) A frame from the Supplementary Movie 1 showing magnetoreceptive e-skin applied onto a wrist of a person and used for interfacing with a smartphone. **An inset shows the magnified image of the magnetoreceptor and conductive paint contact points.** The interaction with e-skin is realized relying on the magnetosensitive skin, the recognized input is transferred to the smartphone. (b-d) A series of reconstructed EMRT interactive segments shown in the Supplementary Movie 1, suggesting touchless input of incorrect and correct graphical PIN codes to unlock the smartphone. The recorded and interpolated stylus trajectory is shown with a solid orange line. (e-g) Representation of the corresponding snapshots of the smartphone interface.

4. *“The methodology for adhering the film to the skin and whether the adhesive affects performance is not discussed. A brief description of the seamless integration methods onto human skin and how the film maintains its functionality upon application should be provided.”*

Answer: We thank the Reviewer for his/her suggestion. The methodology for adhering the functional film to skin is a common issue in the field of flexible electronics and has been extensively explored. In this work, flexible magnetoresistive films were adhered onto the skin with the assistance of vaseline as an adhesive material. Other adhesives can be also used including porous silicon based elastomers (DOI: 10.1038/ncomms5779), PVA (DOI: 10.1038/nmat3755) and hydrogels (DOI: 10.1126/science.aau0780). T

5. *“The film testing, especially the evaluation of vapor permeability, is well organized. Supplementary Figure 12 shows the film flattening back to 67%; apart from GMR, how is the film’s performance impacted?”*

Answer: The magnetoreceptor’s stretchability comes from its wrinkled structure, which is created when pre-strain is released. The development of these wrinkles could potentially influence the GMR performance, as well as the visual transparency and permeability of the magnetoreceptor. When stretched, the GMR response of the magnetoreceptor exhibits minimal variation, indicating the mechanical robustness provided by the continuous mesh. The transparency and permeability also largely depend on the type of polymeric substrate used alongside the GMR mesh. In addition to the VHB tape utilized in this study, other materials like polydimethylsiloxane (DOI: 10.1002/adma.202003155) and eco-flex (DOI: 10.1002/aelm.201500345; 10.1002/adma.201504659) can act as stretchable polymers, giving mechanical pre-strain to the GMR mesh. As stretched, the polymer substrate’s thickness varies, thereby altering the light transmission through the magnetoreceptors. A thinner polymer substrate generally means more transparency for the magnetoreceptor.

Response letter

We would like to thank the Reviewers for their remarks and suggestions, all of which are addressed in the revised manuscript. All changes in the manuscript are indicated in blue.

Our itemized responses to all the Reviewer's comments are below.

Reviewer #2 (Remarks to the Author):

The authors have sufficiently addressed the comments and, in my opinion, the manuscript should be published.

Answer: We thank the reviewer for his/her positive feedback on the revised version of the manuscript and recommendation for its publication.

Reviewer #3 (Remarks to the Author):

The manuscript entitled 'Scalable magnetoreceptive e-skin for energy-efficient high-resolution interaction towards undisturbed extended reality' introduces an innovative method for integrating magneto-inductive functionality into electronic skin. It addresses key limitations in scalability and functionality of electronic skins, paving the way for advanced human-computer interaction, particularly in the realm of extended reality. The authors have responded comprehensively to the relevant questions, supplemented the manuscript with appropriate experiments, and made substantial revisions. Therefore, I endorse the publication of this paper in Nature Communications.

Answer: We appreciate this positive assessment of the manuscript by the Reviewer. We thank the Reviewer for his/her recommendation to publish the manuscript.

Reviewer #4 (Remarks to the Author):

This manuscript presents a method for creating spatially continuous, high-resolution magnetic field maps using the electrical magnetoresistive tomography (EMRT) technique. This approach is based on an extended magnetoreceptive medium and electrical resistance tomography, eliminating the need for active transistor matrices and intricate sensor setups typically seen in traditional active-matrix systems. The subject of this study is intriguing and holds significant potential.

Answer: We thank the Reviewer for his/her suggestions and comments on the manuscript. In the revised version, we addressed these remarks by providing further discussion as well as by adding new experimental data. We hope that the revised version of the manuscript can be recommended for publication.

However, the manuscript lacks some fundamental information. Based on the reported results, I can recommend this study for possible publication in Nature Communications if the authors address the following comments:

1. The authors did not provide reliable evidence for the practical application of the magnetoreceptive contact lens. This shortcoming needs to be addressed by demonstrating practical applications or providing detailed experimental evidence.

Answer: The experimental results shown in our manuscript validate the possibility to enhance the performance of any device by seamless introduction of magnetoreception. The mechanical flexibility and transparency of the EMRT driven e-skins enables their use in combination with light emitting surfaces or/and visible light detectors. These different application scenarios were demonstrated in the manuscript with distinct demonstrators, e.g., PC screen, smartphone screen, and contact lens.

We envision that magnetoreception can enhance functionality of smart contact lenses by adding a sense of motion detection. Our demonstrator with contact lens showcases one deliberately chosen use case scenario, where the detection of a finger movement is used to manipulate objects in augmented reality.

In our manuscript, we performed detailed analysis of the interaction distance, sensitivity, transparency to justify that this application scenario is technically feasible. We are convinced that this extensive list of characterisations reported in the main text and supporting information already provides a detailed experimental evidence of the possibility to realize magnetoreceptive contact lens, which is in line with the question of the Reviewer.

We note that the requested by the Reviewer practical application of the contact lens in real world settings would require many more steps including CMOS interfacing, wireless data transmission, medical device approval. We sincerely ask the Reviewer for understanding that these tasks are not performed in this study. Indeed, these are lengthy and costly endeavours, which are beyond the scope of the manuscript, which aims to demonstrate large area fabrication of transparent and breathable magnetoreceptors.

2. The magnetoreceptor can only be activated by magnets within a 1 cm range, which may impact the field of view when using an electromagnetic pen over the device grid. The authors should explain this limitation and propose possible practical solutions.

Answer: We thank the Reviewer for pointing out this aspect. This issue can be addressed in two ways. First, the magnetic pen can be replaced with magnetic skins made of mechanically soft magnetic composites, as confirmed by our measurements (please see supplementary figure 7 and related movie). These magnetic skins can be easily attached to fingers, serving as a source of magnetic stimuli and reducing the impact on the field of view caused by the magnetic pen during operation. We note that using magnetic skins on a finger has another advantage: Since two eyes provide stereo vision, partial obstruction of the view of one of the eyes should not cause significant discomfort during interaction with the device.

This remark is added to the manuscript (page 19):

We note that the magnetic pen can be replaced with magnetic skins made of mechanically soft magnetic composites (Supplementary Figure 7). These magnetic skins can be easily attached to fingers, serving as a source of magnetic stimuli and

reducing the impact on the field of view caused by the magnetic pen during operation. Using magnetic skins on a finger has an important advantage: Since two eyes provide stereo vision, partial obstruction of the view of one of the eyes should not cause significant discomfort during interaction with the device.

The mentioned working distance of 10 mm can be enhanced by changing the magnetoreceptive medium [e.g., instead of Co/Cu GMR multilayers it is possible to use more sensitive Py/Cu GMR sensors (please see supplementary figure 13) or even sensors relying on the tunnelling magnetoresistance effects as well as use of more sophisticated data processing algorithms capturing the gradient of the resistance change over the magnetoreceptor area. This information is already available in the manuscript (page 9, first paragraph).

3. The wireless technology behind the contact lens system is not clearly explained. The authors should specify whether they incorporate specific wireless technology and, if so, provide details on its implementation and functionality.

Answer: In our work, we focus on the development of the on-skin magnetoreceptive human machine interfaces that are just one of many components that are required for real life realization of the proposed concept of a smart contact lens. Design and development of such components as wireless information/power transmission from/to such a device is beyond the scope of the manuscript.

Motivated by the remark of the Reviewer, we added a remark on the next steps related to the wireless data transmission (page 19):

Next steps should also include demonstration of wireless communication with the magnetoreceptive contact lenses⁴⁹⁻⁵¹.

49 Huang, X., Liu, Y., Kong, G. et al. Epidermal radio frequency electronics for wireless power transfer. *Microsyst Nanoeng* 2, 16052 (2016). <https://doi.org/10.1038/micronano.2016.52>

50 Khan, S. R. et al., Wireless Power Transfer Techniques for Implantable Medical Devices: A Review. *Sensors* 20, 3487 (2020). <https://doi.org/10.3390/s20123487>

51 Nhuyen, D. H. Optical Wireless Power Transfer for Implanted and Wearable Devices. *Sustainability* 15, 8146 (2023). <https://doi.org/10.3390/su15108146>

4. The authors need to investigate and discuss the biocompatibility of the developed e-skin. It is essential to ensure that all materials used are safe for long-term use on human skin. The manuscript mentions that the magnetoreceptive electronic skin is permeable to vapor, moisture, and sweat, reducing the risk of skin irritation. The authors should provide a detailed analysis and experimental validation of these properties to ensure the material's practical biocompatibility.

Answer: Long-term safe and stable operation of e-skins is ensured by the use of appropriate encapsulation layers that are suitable for the targeted use case scenario. In the case of

magnetoreceptive skins, potential hazards are related to the use of cobalt-based magnetic sensing layers and circulating electric currents in resistive sensors. Hence, the use of insulating encapsulation later is imperative for our sensors when aiming on-skin applications. For encapsulation, we used plastic varnish which suites the purpose of our experiments. However, other encapsulation layers, which are typically used in flexible on-skin electronics like PDMS and ecoflex can be readily applied as well.

Following the remark of the Reviewer, we added the related comment to the revised manuscript (page 23-24):

Long-term safe and stable operation of e-skins is ensured by the use of appropriate encapsulation layers that are suitable for the targeted use case scenario. In the case of magnetoreceptive skins, potential hazards are related to the use of cobalt-based magnetic sensing layers and circulating electric currents in resistive sensors. Hence, the use of insulating encapsulation later is imperative for our sensors when aiming on-skin applications. For encapsulation, we used plastic varnish (Plastik70, CRC Industries UK Ltd. United Kingdom). However, other encapsulation layers, which are typically used in flexible on-skin electronics like PDMS and ecoflex can be readily applied as well.

5. There is insufficient information on the tactile sensitivity of the e-skin. The authors should provide data on pressure sensitivity, range, and limit of detection (LOD). Additionally, they should include $\Delta R/R_0$ versus applied pressure and demonstrate the tactile response for cyclic applications, with a minimum of 1000 cycles.

Answer: The large-area magnetosensitive e-skin presented in our manuscript is designed to work in a *contactless* regime relying on the interaction with magnetic stray fields. Therefore, the device is designed not to provide pressure readings and can be implemented into integrated e-skin devices as auxiliary interaction channel to any surface or device. The insensitivity of the magnetoreceptive GMR mesh to the pressure and mechanical deformation was demonstrated in the Supplementary Movie 6, where the sensor responds only when it is approached by fingers fashioned with magnetic skin and does not show any response when pressed by bare fingers. Additionally, the mechanical stability of the GMR mesh upon mechanical stimuli is shown by bending and stretching experiments shown in the Supplementary Figures 16-18.

We note that electrical impedance tomography is already widely used to realize touch and pressure sensors. The relevant references are reported in our manuscript as:

[38] Park, K. *et al.* A biomimetic elastomeric robot skin using electrical impedance and acoustic tomography for tactile sensing. *Sci. Robot.* **7**, eabm7187 (2022).

[40] Silvera-Tawil, D., Rye, D., Soleimani, M. & Velonaki, M. Electrical impedance tomography for artificial sensitive robotic skin: A review. *IEEE Sens. J.* **15**, 2001–2016 (2015).

Our touchless on-skin sensor technology may be readily combined with these pressure sensitive devices to realize a multifunctional sensor.

Stimulated by the remark of the Reviewer, we added the following comment to the manuscript (page 16):

The large-area magnetosensitive e-skin presented in our manuscript is designed to work in a *contactless* regime relying on the interaction with magnetic stray fields. Therefore, the device is designed not to provide pressure readings. The insensitivity of the magnetoreceptive GMR mesh to the pressure and mechanical deformation is demonstrated in Supplementary Movie 6, where the sensor responds only when it is approached by fingers fashioned with magnetic skin and does not show any response when pressed by bare fingers. Additionally, the mechanical stability of the GMR mesh upon mechanical stimuli is shown by bending and stretching experiments shown in the Supplementary Figures 16-18. We note that electrical impedance tomography is already widely used to realize touch and pressure sensors^{38,40}. Our touchless on-skin sensor technology may be readily combined with these pressure sensitive devices to realize a multifunctional sensor with touchless and tactile sensitivity²².

6. The reliable performance of the e-skin under different humidity and temperature conditions is unclear. The authors should investigate and report the effects of temperature, pH, and humidity on the e-skin's performance.

Answer: To address these comments of the Reviewer, we performed a series of additional experiments to investigate the stability of the GMR mesh under different environmental conditions. We note that these experiments are carried out with an encapsulated sensor to prevent electrical shortage upon immersion into conducting liquids. Two sets of experiments have been performed: change of temperature in the range from 10 to 60°C (typical operation temperature range for on-skin devices) and when submerged into liquids with different pH (mimicking exposure to physiological liquids, like sweat, etc.). The experimental results are shown in figures below and are introduced to supplementary materials as Supplementary Figures 21 and 22, respectively. With increasing temperature the base resistance (R_0) of the GMR mesh increases, which is typical for metals. This resistance increase leads to a slight decrease in the GMR (less than 5%). The change of the sensor resistance and MR performance is reversible. After the sensor is cooled down to room temperature, it recovers its initial base resistance level and MR performance. The exposure of GMR mesh to basic and acidic solutions (with tested pH from 1 to 10) does not affect its magnetoreceptive performance since the encapsulation layer protects the metallic components of the device from damage. These experiments carried out by measuring the sensor performance when it is submerged in liquids indicates that our large area magnetoreceptive skins can operate in a humid environment.

Motivated by this comment of the Reviewer, the following remark is added to the manuscript (page 23) as well as new supporting figures 21 and 22:

In particular, the encapsulated sensors operate in a broad temperature range from 10 to 60°C (typical operation temperature range for on-skin devices; Supplementary Fig. 21) and can be submerged into liquids with different pH (mimicking exposure to physiological liquids, like sweat, etc.; Supplementary Fig. 22).

Supplementary Figure 21. Temperature stability of the transparent GMR mesh. The mesh sensor was placed onto a Peltier element and the device temperature was monitored using infrared camera. (a) The GMR curves measured of the mesh sensor at different temperatures. (b) The temperature variation of the base resistance (R_0) and GMR response of the sensor.

Supplementary Figure 22. The stability of the GMR mesh sensor upon immersion into media with different pH. (a) The GMR curves measured of the mesh sensors upon immersion into liquid media with different pH: citric acid (pH =1), vinegar (pH = 3), DI water (pH = 7), baking soda solution (pH = 9), soap solution (pH = 10). (b) The magnitude of GMR response of the encapsulated mesh sensor submerged into solutions with different pH.

7. Have the authors conducted long-term performance tests to assess the durability and stability of the magnetoreceptive electronic skin? It is important to determine if the sensor can accurately detect and localize magnetic input when deformed or stretched over time.

Answer: We performed detailed characterization of the sensor stability. Namely, static bending tests of flexible sensors to different bending radii are shown in Figure 3c of the main text (also supplementary figure 16). Tensile testing of elastic sensors is summarised in Figure 3d of the main text (also supplementary figures 17 and 18). During the preparation of demonstrators featuring interactions with a smartphone (figure 1d, supplementary figure 9 and related videos) and immersive VR (figure 4 and related videos), the sensors were many times mounted on skin, removed from skin and placed again on skin at different locations. Even after these harsh manipulations during on-skin experiments, the sensors did not reveal degradation of their performance. We note that the stability of the sensors is also given by the fact that all reported results in the initially submitted manuscript (March 2024), first revised manuscript (June 2024) and second revised manuscript (current submission; October 2024) are performed on the same family of sensors. Our measurements performed over more than half a year, which are added to the manuscript upon each of the revision rounds do not show any sign of degradation of the sensors and their performance to detect magnetic fields and to localise magnetic stylus.

Response letter

Reviewer #4

The manuscript entitled "Scalable magnetoreceptive e-skin for energy-efficient high-resolution interaction towards undisturbed extended reality (NCOMMS-24-15778C)" is interesting, and the quality of the manuscript is sufficient to be considered for publication. Overall, the authors have addressed my comments in the revised manuscript, and I recommend it for possible publication in Nature Communications.

We appreciate this positive statement of the Reviewer and his/her recommendation to publish the manuscript in Nature Communications.